

# Four decades of global surface albedo estimates in the third edition of the CLARA climate data record

Aku Riihelä[1], Emmihenna Jääskeläinen[1], Viivi Kallio-Myers[1]

[1]Finnish Meteorological Institute, Helsinki, FI-00560, Finland

*Correspondence to*: Aku Riihelä (aku.riihela@fmi.fi)

**Abstract.** We present the surface albedo data in the third edition of the CM SAF cLoud, Albedo and surface Radiation (CLARA) data record family. The temporal coverage of this edition is extended from 1979 until near-present day. The core algorithms and data format remain unchanged from previous editions, but now also white- and blue-sky albedo estimates are available for the first time in CLARA data. We present an overview of the retrieval, followed by an assessment of the

accuracy and stability of the data record, based on collocated comparisons with reference surface albedo measurements and intercomparisons with preceding satellite-based albedo data records. Specific attention is paid to addressing the spatial representativeness problem inherent in the point-to-pixel validation of satellite-based coarse surface albedo estimates against *in situ* measurements. We find the CLARA-A3 albedo data to match or improve upon the accuracy and robustness of the predecessor record (CLARA-A2), with good agreement found when compared to *in situ* measurements. In cases of a large

bias, spatial representativeness of the measurement site typically explains most of the increase. We conclude with a summarizing discussion on the observed strengths and weaknesses of the new data record, including guidance for potential users. The data is available through DOI: 10.5676/EUM_SAF_CM/CLARA_AVHRR/V003.

## 1 Introduction

Energy from solar radiation is the principal energy source for Earth's climate and its ecosystems. At the surface level, the up-

and downwelling (shortwave) solar fluxes and the thermal (longwave) radiative fluxes combine to form the surface radiative energy budget (SRB), a key component of the climate system. Surface albedo ($\alpha$) determines the fraction of the incident shortwave solar flux which is reflected away from the Earth's surface and is therefore an important driver of SRB. Here, we define albedo values as fractions between 0 and 1. Surface albedo magnitude is determined not only by the radiative properties of the surface over the examined wavelengths, but also the directionality of the incoming solar flux. Snow and sea

ice are among the brightest natural surfaces. This also implies that should they melt away, they are always replaced by much darker soil and open water, giving rise to the snow and ice albedo feedback (SIAF) where increasing heat absorption continues to feed additional melting (Budyko, 1969; Sellers, 1969). However, it is equally important to note that vegetation too drives changes in albedo and reacts to them (e.g. Tian et al., 2014; Beringer et al., 2005).



It is therefore clear that the surface albedo of the Earth should be continuously monitored, particularly over the remote
cryospheric domains where gains and losses in snow and ice may exert substantial climatic influence. In practice, efficient
and continuous monitoring of surface albedo at global scale requires satellite observations. For applicability in climate-
related studies, it is further required that the observations have multidecadal coverage, are carefully intercalibrated across
sensor specimens, are processed into albedo estimates with consistent algorithms and auxiliary input data and are finally
carefully validated against reference observations to determine their quality.

To answer this need, the Satellite Application Facility on Climate Monitoring (CM SAF), a project of the European
Organization for the Exploitation of Meteorological Satellites (EUMETSAT), produces and distributes decadal scale Climate
Data Records (CDR) from the longest continuous satellite data records available. Here, we present the third edition of global
surface albedo data in the CLARA (CM SAF cLoud, Albedo and surface Radiation) data record family. The CLARA-A3
(Karlsson et al., 2023) is based on four decades of intercalibrated satellite observations from the AVHRR (Advanced Very
High Resolution Radiometer) optical imagers, with full global coverage. The data record is produced and delivered in two
components: The CDR which covers 1979-2020, and an Interim Climate Data Record (ICDR), which provides continuous
updates to the CLARA-A3 record from 2021 until near-present day. The essential description of the data record as a whole is
available in Karlsson et al. (2023) and is not repeated here. Here we instead focus on the surface albedo component of
CLARA-A3. We begin by introducing the algorithms for the derivation of surface albedo estimates. Then, the performance
of the data is evaluated against reference in situ observations, the stability and uncertainty of the data record is discussed, and
finally, we provide an intercomparison of the new data against the widely used MCD43 (Edition 6.1) and the predecessor
CLARA-A2 data records. The results are then summarized in a discussion on the observed strengths and weaknesses of the
new surface albedo data record.

## 2 Data record description and algorithm overview

The surface albedo estimates in the CLARA-A3 climate data record (CDR) are available as five-day (pentad) or monthly
means between 01/1979 and 12/2020, with a continuation through an Interim Climate Data Record (ICDR). All observations
are from different members of the Advanced Very High Resolution Radiometer (AVHRR) spaceborne optical imager
family; the observed radiances have been intercalibrated to eliminate inter-sensor jumps in the data record (after Heidinger et
al., 2010). This preprocessing step is crucial for enabling climate trend studies for the derived geophysical variables. The
data are provided on a global 0.25 degree lat-lon grid, with the polar regions also covered by subsets on a 25 km resolution
equal-area EASE2 grid. The core algorithm closely follows that of the predecessor records (Riihelä et al., 2013).
Specifically, it is a sequential progression through topography and atmospheric corrections, Bidirectional Reflectance
Distibution Function (BRDF) treatments, and the narrow-to-broadband conversion to shortwave broadband surface albedo.
However, important expansions in scope and coverage are now available for the first time.

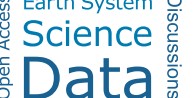

The primary change is that in addition to the estimates for Directional-Hemispherical Reflectance (DHR; also black-sky albedo) which formed the content of the predecessor CLARA albedo data, we now provide also the estimates for Bidirectional Reflectance under fully diffuse illumination (BHR$_{ISO}$; also white-sky albedo) and the Bidirectional Reflectance under ambient illumination conditions (BHR; also blue-sky albedo). These variables are henceforth called SAL, WAL, and BAL, respectively. Overviews of their retrieval process are available in the following subsections, with a complete

description of all algorithm details available in the Algorithm Theoretical Basis Document (ATBD) through the data record's DOI. For readers familiar with previous versions of the data record, a summary of changes in input and retrieval algorithms between CLARA-A3 and CLARA-A2 albedo data is shown in Table 1.

The data are provided in NetCDF-4 files, compliant with the Climate and Forecast (CF) metadata conventions (v1.7) as well as the NetCDF Attribute Convention for Data record Discovery (v.1.3).

**Table 1: Summary of algorithm and input changes between CLARA-A3 and A2 climate data records for surface albedo**

| Retrieval algorithms | CLARA-A2 | CLARA-A3 | Comments |
|---|---|---|---|
| WAL over snow-free land | None | Yang et al. (2008) | |
| WAL over snow and ice | None | Manninen et al. (2019) | |
| SAL retrieval using cloud probability data | None; based on binary cloud mask | Manninen et al. (2022) | Fixed threshold of <20% cloud probability for retrieval |
| Sun Zenith Angle (SZA) normalization | Only ocean surfaces normalized to SZA of 60 deg. | No normalization applied for any surfaces | Mean SZA available in data for users' own normalization |
| BAL over all surfaces | None | Direct illumination-weighted mean of SAL and WAL | Fraction of direct illumination estimated during SAL atmospheric correction; details in ATBD |
| Retrieval inputs | | | |
| Atmospheric composition (ozone, water vapour, surface pressure) | ERA-Interim | ERA5 for CDR; ERA5T for ICDR | |
| AVHRR radiance (inter)calibration | Karlsson et al. (2017), originally after Heidinger et al. (2010) | Pygac 1.6.0, originally after Heidinger et al. (2010) | |
| Atmospheric correction coefficients | Always continental | Desert coefficients over barren terrain, otherwise continental | |



Figure 1a shows an example of the monthly mean BAL from April 2015. Insufficient solar illumination prevents retrievals over Antarctica and near the North Pole, whereas the albedo of the Arctic sea ice in general is at or near its seasonal maximum. Figure 1b shows the zonal means of SAL, WAL, and BAL for the month in question. Over the margins of the
seasonal snow cover zone, Figure 1 shows the mean of snowy and snow-free BAL, weighted by respective sampling. For SAL, separate data layers are provided for snow/ice and snow-free albedo estimates. The principal variables of interest to users for SAL, WAL, and BAL are the combined data layers for mean surface albedo (identifier "*_all*" in data files), where snow and snow-free observations are combined as a weighted mean by counts of snow and snow-free observations (also provided in the data files).


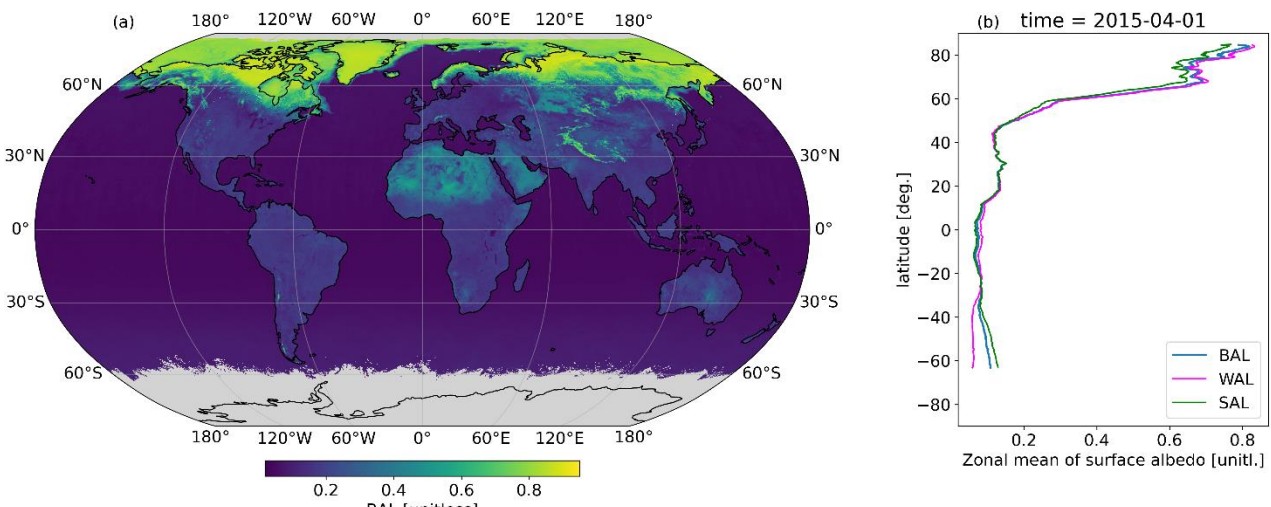

**Figure 1: (a) an example of the monthly mean blue-sky (BAL) surface albedo from April 2015. (b) the zonal means of the blue-sky (blue, thick), white-sky (WAL, orange), and black-sky (SAL, green) surface albedo.**

## 2.1 Algorithm overview for black-sky albedo (SAL)

AVHRR channels 1 and 2, 0.58-0.68 μm (CH1) and 0.725-1 μm (CH2), respectively, are the radiance sources for SAL generation. The algorithm flow is outlined in Figure 2. The first step towards the estimation of black-sky albedo is the identification and exclusion of cloudy and cloud-contaminated areas in the AVHRR imagery. The CLARA-A3 record provides probabilities of cloudiness for each imaged AVHRR pixel, obtained through Bayesian classification (Karlsson et al., 2020). Choosing a threshold of cloud probability (CP) for discarding the observations is a trade-off between sampling
density and robustness; extensive tests indicated that a universally applied threshold of 20% cloud probability provides the best balance for SAL retrieval (Manninen et al., 2022). Furthermore, all observations with an unfavourable illumination/observation geometry (Sun Zenith Angle >70 deg. or Viewing Zenith Angle >60 deg.) are discarded. A flag for snow-covered terrain and sea ice (verified with OSI SAF sea ice concentration data) is also provided from the cloud processing (Karlsson et al., 2023).

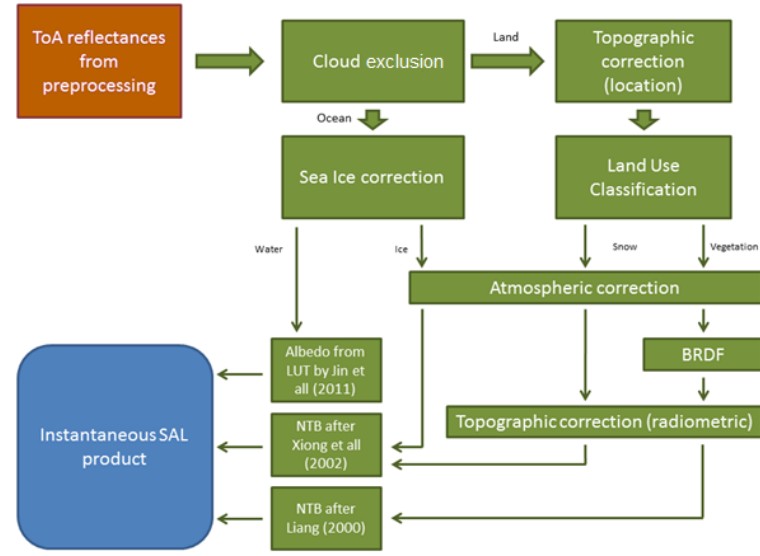


**Figure 2: Flowchart of level 2 processing for black-sky albedo (SAL) in CLARA-A3.**

The topography correction for geolocation and radiometry in AVHRR imagery is applied exactly as in the predecessor CLARA records. As AVHRR geolocation is calculated on a geodesic reference ellipsoid, a combination of sufficiently large elevation and viewing angle requires across-track shifting of pixels to obtain true geolocation. Over sufficiently rugged
terrain (mean slope larger than 5 degrees, i.e. mountainous areas), further calculations attempt to improve the BRDF correction in image radiometry (atmospherically corrected surface reflectances) by accounting for the effects of view- or illumination-shadowed subslopes (Manninen et al., 2011), as calculated from DEM and imaging geometry, in each AVHRR pixel to be corrected. Data sources are the Shuttle Radar Topography Mission (SRTM) Digital Elevation Model (DEM) for latitudes between +/- 60°, and the GTOPO30 DEM elsewhere.

For the atmospheric correction necessary to reduce the satellite-observed Top-of-Atmosphere (ToA) reflectances to surface reflectances, we continue to apply the Simplified Method for Atmospheric Correction (SMAC; Rahman and Dedieu, 1993). The principal inputs required to describe the atmospheric composition are total column water vapour content, ozone content, surface pressure, and the aerosol optical depth (AOD) of the atmosphere at 550 nm wavelength. In CLARA-A3 CDR, the data source for water vapour, ozone, and surface pressure was updated to the ERA5 atmospheric reanalysis (Hersbach et al.,
110  2020).

For AOD over land surfaces, we continue to use the time series developed from TOMS and OMI Aerosol Index (AI) observations (Jääskeläinen et al., 2017). Owing to caution related to potentially increased uncertainty in the UV-waveband observations of OMI (Kleipool et al., 2022), AOD for years 2015-2020 was treated as a day-of-year climatology based on data from 2005-2014. Furthermore, as SMAC has limitations in accuracy with high aerosol loading conditions, all AVHRR
observations where the assigned (daily) AOD exceeds 1.0 are discarded. Over snow, ice, and water the fixed AOD of 0.1



was updated to 0.05 to better match common aerosol loading conditions over the polar regions during summer (Tomasi et al., 2012). We acknowledge that these choices hinder retrieval accuracy during rapidly changing aerosol loading conditions. However, as our temporal resolution is either five days or one month, the capacity for rapid change tracking is in any case only partial.

At this stage, the processing diverges for snow/ice and snow-free land surfaces. We first consider snow-free land. The BRDF correction and conversion to narrowband surface albedos for AVHRR CH1 and CH2 continue to follow the kernel-based approach of Wu et al. (1995) and Roujean et al. (1992). The narrow-to-broadband conversion (NTBC) algorithm also follows Liang et al. (2000). The BRDF correction magnitude is land-cover specific. Dominant land cover for each AVHRR observation is taken from a variety of land cover datasets: The USGS land cover for 1979-1997, GLC2000 for 1997-2002,

GLOBCOVER2005 for 2002-2007, GLOBCOVER2009 for 2007-2012, and ESA LU CCI after 2012. Prior to use, the land cover data are mapped into coarse land cover archetypes (e.g. forest, grassland, etc.) to match the granularity of the BRDF model and to improve inconsistencies during shifts from one land cover data source to another.

Over snow and ice, the retrieval does not attempt a correction for BRDF effects in the level 2 (single swath) processing. Instead, the atmospherically corrected surface reflectances are converted to broadband snow/ice reflectances following

Xiong et al. (2002), noting that the NTBC algorithm also self-adapts to wet and dry snow/ice conditions. Then, the broadband reflectances are aggregated and averaged during level 3 processing, relying on dense angular sampling of the AVHRR sensor to cover the angular domains most relevant for bidirectional reflectance variation for snow and ice. There are two principal justifications for this choice. First, as seen in Figure 3a, available clear-sky AVHRR observations over the polar regions (here a site on the Greenland Ice Sheet) cover the majority of viewing hemisphere during the summer months.

Keeping in mind that the reflectance signature of snow and ice is symmetric about the principal plane, we can see that the unsampled part of the viewing hemisphere is a relatively narrow angular domain about the cross-principal plane. In Figure 3b, an illustration of angular variability in snow reflectance after recent modelling efforts by Jiao et al. (2019) confirms that this angular range contributes little to the overall angular reflectance signature of snow.

The second part of the justification for this simple method lies in the lack of universally applicable BRDF models valid for

all naturally occurring snow and (sea) ice conditions. While clear progress has been made in BRDF treatments and albedo retrievals of optically deep snow cover (e.g. Jiao et al., 2019; Kokhanovsky et al., 2019, 2021) as well as sea ice (Malinka et al, 2016; Pohl et al., 2020), the available methods either require *a priori* information or assumptions about the state of the snow/ice (e.g. sufficient depth of snow) or are designed for more modern, higher resolution optical spaceborne sensors such as MERIS or MODIS. While we acknowledge that the method chosen for CLARA retrievals cannot match the precision of

specifically designed algorithms, we maintain that the snow and sea ice albedo estimates in CLARA-A3 are always based on realized AVHRR reflectances and will avoid retrieval errors resulting from choosing an inappropriate snow/ice BRDF model for the scene. This is particularly relevant for sea ice, where the surface conditions may be a superposition of wet & dry snow, bare ice and the surface scattering layer (e.g. Smith et al., 2022), melt ponds, and open water and leads.

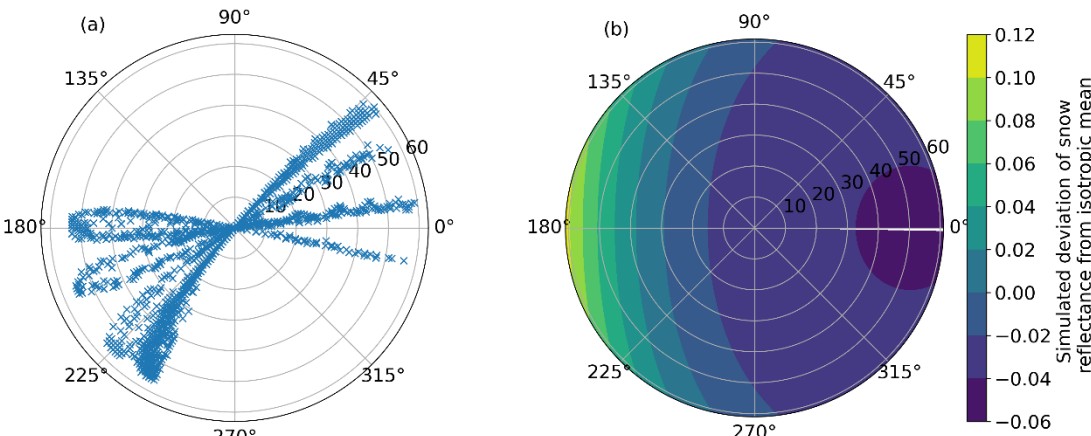

**Figure 3: (a) Satellite zenith and relative azimuth angles of successful CLARA-A3 SAL retrievals during 2020 at Summit Camp in Greenland. The polar plot shows viewing (satellite) zenith angles in the radial axis and relative azimuth angles in the angular axis. (b) An example of the deviation of angular snow reflectance from its isotropic mean as modeled by the snow kernel proposed by Jiao et al. (2019) in the same polar coordinates.**

For water bodies, surface albedo is derived following Jin et al. (2011). Surface albedo for both direct and diffuse illumination is calculated as the sum of surface and ocean volume contributions, although the volume component is set to a fixed value of 0.006 (Jin et al., 2011). The surface contribution depends on the surface roughness of the ocean surface, which in turn is driven by wind speed, and Sun Zenith Angle (for black-sky albedo). A correction for foam is applied, dependent on the fractional coverage of white-caps in the area of interest. Wind speeds over global oceans are compiled from observations of the Scanning Multi-channel Microwave Radiometer (SMMR), Special Sensor Microwave Imager (SSM/I), Special Sensor Microwave Imager Sounder (SSMIS) instrument series, supplemented with anemometer-based wind speed data (Tokinaga, 2013) where needed. During the SMMR era of 1979-1984, wind speeds are available as monthly means (Wentz,1997, Vazquez, 1997). Anemometer observations and climatological values fill the gap from January 1985 – June 1987 at monthly mean resolution, after which SSM/I and SSMIS observations are available on a daily basis, although here we aggregate the data to match the temporal resolution of our pentad means. Full details of these calculations are available in the ATBD.

Note that the AVHRR-observed reflectances are not used in the estimation of ocean surface albedo. The choice is justified by the relatively uniform behaviour of ocean albedo as a function of wind speed (in absence of large chlorophyll concentrations, for which no global observational data exists reaching back to 1979) and the marked reduction in computational needs, gained when deriving ocean albedo through this parameterized approach.

At this stage the estimation of black-sky albedo at the overpass/swath level is complete. The data are then transferred to the level 3 aggregation code, which first calculates statistical parameters of SAL (e.g. moments, standard deviation, skewness and kurtosis) in each 0.25 deg or 25 km grid cell. The black-sky albedo parameters are finally corrected for effects of non-zero cloud probability (Manninen et al., 2022) and recorded as the spatiotemporal mean provided in the product files.



## 2.2 Algorithm overview for white-sky albedo (WAL)

The white-sky albedo for snow-free land surfaces is derived from the estimated black-sky albedo and SZA following Yang et

175 al. (2008):

$$WAL = \alpha_{white} = \frac{1 + 1.48 \; \cos \theta_z}{2.14} \alpha_{black} \tag{1}$$

where $\alpha_{black}$ is the black-sky albedo (SAL) and $\theta_z$ the solar zenith angle. WAL over snow-free land is calculated during level

2 processing and averaged to form the pentad and monthly means.

For snow-covered land and sea ice, WAL is estimated based on statistical relationships of black-sky and white-sky albedo

parameters as observed in *in situ* measurements (Manninen et al., 2019). Only the temporal mean of white-sky albedo is

derived, and the applied equation is:

$$\text{WAL} = \overline{\alpha_{black}} \cdot$$

$$[1 + \overline{\theta_z}(1.003 + 0.128 \, \overline{\theta_z} - 1.390 \, \overline{\alpha_{black}} + 0.0341 \widetilde{\alpha_{black}} - 0.998 \, \sigma_{black} - 0.0155 \, \gamma - 0.000625 \, \beta)] \tag{2}$$

where $\overline{\alpha_{black}}$ is the temporal mean of black-sky albedo (SAL) and $\widetilde{\alpha_{black}}$ refers to the median, $\sigma_{black}$ to the standard

deviation, $\gamma$ to the skewness and $\beta$ to the kurtosis of the black-sky albedo distribution, and $\overline{\theta_z}$ is the mean SZA of the period

in radians.

The observed empirical relationships change with the presence of above-snow vegetation, therefore the following equation is

applied for forested snow-covered areas (Manninen et al., 2019):

$$WAL = \overline{\alpha_{black}} \cdot$$
$$[1 + \overline{\theta_z}(-0.592 + 0.709 \, \overline{\theta_z} - 11.4 \, \overline{\alpha_{black}} + 11.0 \widetilde{\alpha_{black}} + 5.10 \, \sigma_{black} + 0.0204 \, \gamma - 0.0205 \, \beta)] \tag{3}$$

Snow-covered forests are identified from land cover and the snow cover flag. However, in sparse (boreal) forests the scene

reflectance may still be too bright for the equation to be applicable. Therefore, the equation for snow-covered forest is

applied only if the observed SAL is less than 0.5.

Finally, testing during the CDR processing indicated that the empirical nature of WAL retrieval led to a slight

underestimation over the brightest snow surfaces, but also overestimation over sea ice where the statistical parameter

distributions differed from snow. Therefore, WAL over the brightest snow was bias corrected with a multiplication factor of

$\exp(0.1 * WAL^4)$, and WAL over sea ice was constrained to not exceed SAL by more than 10% relative, consistent with

results from prior studies (Key et al., 2001). Further details on WAL derivation are available in the ATBD.

## 2.3 Algorithm overview for blue-sky albedo (BAL)

The blue-sky surface albedo is estimated as (Lucht et al., 2000; Pinty et al., 2005; Schaepman-Strub et al., 2006; Román et

al., 2010):

$$\alpha_{blue} = f_{dir}\alpha_{black} + f_{diff} \, \alpha_{white} = f_{dir}SAL + f_{diff} \, WAL \tag{4}$$





where $f_{dir}$ and $f_{diff}$ are the fractions of direct and diffuse irradiance, respectively, so that $f_{dir} + f_{diff} = 1$. The equation requires simplifying assumptions about the properties of the incoming diffuse irradiance (Lucht et al., 2000; Pinty et al., 2005), leading typically to underestimations with high SZA. However, given the conservative cut-off of 70 degrees for SZA in CLARA-A3, use of this equation is justifiable.

Both $f_{dir}$ and $f_{diff}$ must be estimated for all AVHRR-imaged scenes during daytime (whether clear or cloudy) in order to obtain realistic temporal means of real-world sky conditions for each aggregation period in question. SMAC-based estimates for these fractions are sufficiently accurate in clear-sky conditions, but for cloudy conditions we estimate them based on an observed sigmoid relationship between $f_{diff}$ and the clearness index (Hofmann and Seckmeyer, 2017). We approximate clearness index with cloud probability (CP), obtaining the following parameterization for $f_{dir}$:

$$f_{dir}(\theta_z, CP) = \frac{f_{dir}(\theta_z, CP=0)}{1 + \exp(0.0919*CP - 4.5951)} = \frac{\exp(-0.1)\cos(\theta_z)}{1 + \exp(0.0919*CP - 4.5951)} \tag{5}$$

The parameterization also depends on SZA ($\theta_z$) so that the $f_{dir}$ variability in clear-sky conditions (CP=0) is realistic.

## 2.4 Data quality and uncertainty indicators

As CLARA-A3 albedo retrieval approach is deterministic rather than probabilistic, the data do not contain direct grid cell-
specific error estimates. However, a wide array of data layers describing data quality are available. Here we highlight the three most important indicators of quality: sampling density, skewness, and kurtosis.

Figure 4a illustrates sampling density in the CLARA-A3 surface albedo CDR through the monthly mean of valid clear-sky GAC-resolution observations in each 0.25-degree grid cell at global level. Variability in sampling at global level relates primarily to changes in the AVHRR constellation; throughout the 1980s and 1990s, typically only one or two sensors were
operational at any given time. In the 2000s, additions of secondary NOAA satellites in the morning and afternoon orbits and the launches of the AVHRR-carrying Metop weather satellites from 2006 onwards have substantially increased the available sampling.

Given the reliance on dense sampling in the CLARA albedo estimates over snow and ice, the issue is particularly relevant over the polar regions. Figure 4b shows mean GAC-resolution observation count over a small area of the Greenland Ice
Sheet for each month in the CLARA-A3 CDR where retrievals were possible given the prevailing solar geometry (marker color). Between early spring/late autumn and midsummer, available sampling may change by a factor of 50. With low sampling, the angular coverage degrades, leading to larger biases in the albedo estimates.

We highlight this behaviour in Figure 5, which displays the deviation of monthly mean BAL from expected climatological surface albedos as a function of sampling and solar geometry over two homogeneous and flat areas: in (a), a dry-snow region
of the Greenland ice sheet, in (b) a cropland/grassland region over central United States. The climatological albedos are set to 0.85 for dry snow (Konzelmann and Ohmura, 1995), and 0.2 for a grassland-cropland mixture (He et al., 2014), and are adjusted for SZA variation following Briegleb et al. (1986). Given months of high sampling, BAL agrees very well with

expectations. Over the ice sheet, the largest deviations occur during spring and autumn where increased uncertainty in the atmospheric correction combines with solar geometry cutoffs in sampling, leading to a low amount of available observations

and therefore higher uncertainty. Most prior Arctic albedo studies using the predecessor CLARA records have only used data from May-August for this reason; the recommendation continues to hold for CLARA-A3. Figure 5 does not show four individual months over the ice sheet where corrupted AVHRR data caused clearly erroneous retrievals. Their effect is discussed in section 5 Discussion, strengths and limitations of the CLARA data.

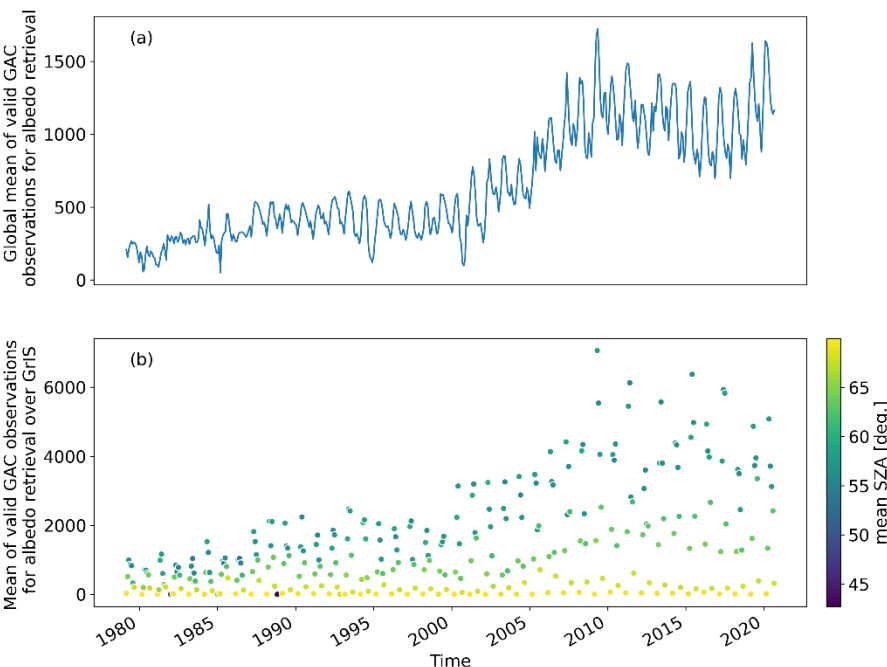


**Figure 4: (a) The global mean of valid clear-sky GAC-resolution AVHRR observations (NOBS) in monthly mean SAL. (b) The amount of valid observations over a small region in the central part of the Greenland Ice Sheet (72-74° N, 39-41° W). Each marker represents a monthly mean, color-coded by the mean Sun Zenith Angle (SZA) of the observations.**

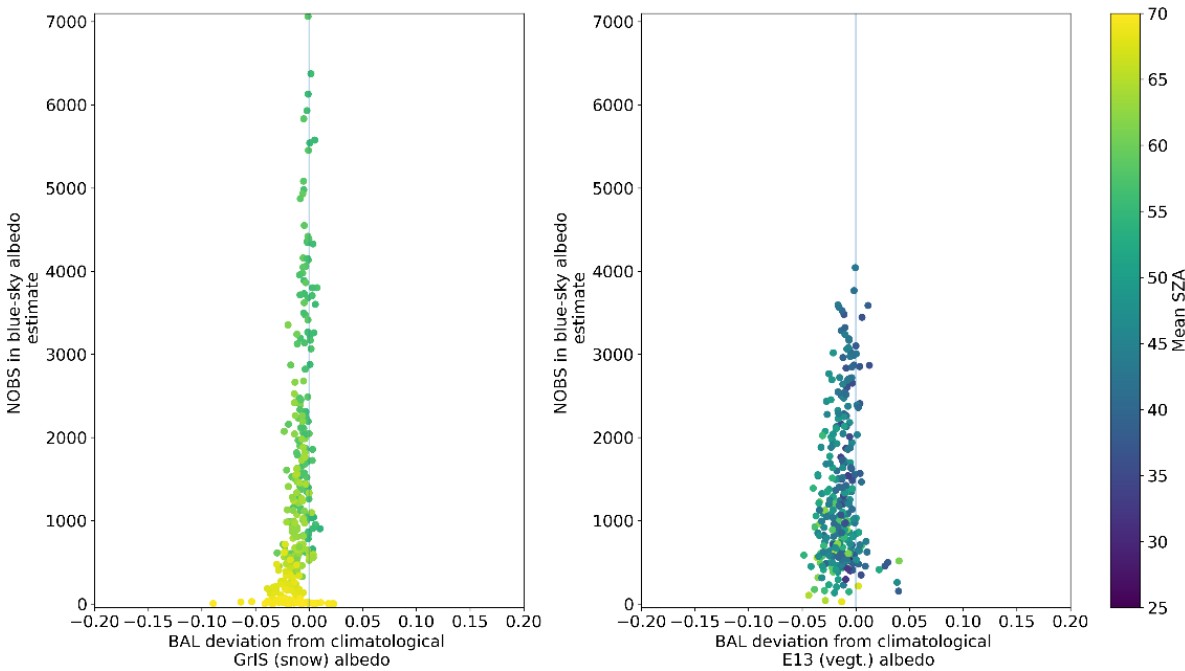

**Figure 5: (a) Deviation of monthly mean BAL from climatologically expected surface albedos over central GrIS and the Kansas plains containing the E13 BSRN site (b) as a function of sampling count and mean SZA (marker color).**

Results from a similar analysis for skewness and kurtosis over the same sites are shown in Supplementary Figure S1. Skewness describes the deviation of data from a normal distribution, both in direction (tails on left or right of distribution center) and the magnitude of the deviation. Kurtosis is a measure for the length of the distribution tails – only deviations more than one standard deviation away from normal distribution center contribute significantly to kurtosis. We see that very large skewness and kurtosis values are possible for the ice sheet area during months with mean SZA close to cutoff and low sampling. Alongside low sampling counts, very large kurtosis and skewness are clearly indicators of low confidence in retrieval robustness. It is recommended that they should be examined and screened before using the data. Note that over snow and ice, skewness and kurtosis are available only in the black-sky SAL data.

Over vegetated surfaces, very large skewness or kurtosis are generally not present. However, especially over high-latitude land surfaces, persistent cloudiness and/or low sun elevation may create conditions of very low sampling and large skewness/kurtosis. This may occur in the pentad means where the aggregation period is shorter.

## 2.5 The Interim Climate Data Record (ICDR) of CLARA-A3 surface albedos

The CLARA-A3 CDR is continued from 2021 to near-present day as an Interim Climate Data Record, retrieved with the same algorithms as the core CDR and delivered in the same format. However, due to timeliness constraints, the atmospheric composition data used in SAL processing is changed from ERA5 to the continuously updated ERA5T reanalysis. AOD and



ocean wind speed data are applied as daily climatologies, and the sea ice concentration data used to sanity-check cloud screening over sea ice regions is changed to OSI-401b. Additionally, the AVHRR radiances in ICDR have not gone through
the same level of intercalibration as the CDR, which is more stable apart from some issues in 2019 and 2020. (please see the Discussion section for details). A 6-month overlap period cross-check between CDR and ICDR suggested that differences in surface albedo of 0.01 – 0.04 are common, with larger differences possible for the brightest snow over ice sheets. This behaviour points to the radiance calibration differences as the likeliest source of discrepancy. We note that the global mean difference in albedo between CDR and ICDR was within 0.01 during the examined period.

## 275 3 Validation of CLARA-A3 surface albedo estimates against reference in situ measurements

Prior to release, the CLARA-A3 surface albedo CDR was validated against decade-spanning *in situ* measurements from the Baseline Surface Radiation Network (BSRN; Driemel et al., 2018), Programme for Monitoring of the Greenland Ice Sheet (PROMICE; Fausto et al., 2021), as well as data from Tara-Arctic (Vihma et al., 2008) and Surface Heat Budget of the Arctic Ocean (SHEBA; Perovich et al., 2002) drifting ice camps in the Arctic Ocean. Tables Table 2 and Table 3 list the
locations of BSRN and PROMICE sites and their periods of coverage. A separate Validation Report (VR) containing full details and results of the effort is available through the data record DOI. Here we present a summary of the most relevant findings, focusing on evaluation at the monthly mean time resolution; pentad mean performance is detailed in Supplementary Material. Retrieval accuracy is quantified through three metrics: mean relative bias (MBE in %), bias-corrected RMS error as a measure of precision (bc-rms, unitless), and the decadal stability of bias, i.e. the temporal trend in
bias as per cent per decade.

For the validation of white-sky (WAL) and black-sky (SAL) estimates, the BSRN in situ measurements were filtered for the amounts of diffuse/direct incident solar flux, requiring more than 98% diffuse/direct irradiance for the white/black-sky validation, respectively. As PROMICE sites do not record the irradiance components, the division was instead made on observed cloud cover, with >0.99 and 0.0 as the requirements for white/black-sky conditions. Strictly speaking, full or non-
existent cloud cover do not necessarily equate to perfect diffuse or direct illumination conditions, thus residual uncertainty in the comparativeness remains for WAL and SAL.

**Table 2: BSRN sites serving as reference surface albedo data sources. Sites assessed as spatially unrepresentative at CLARA resolution are written in red italics.**

| Station code | Name | Latitude (°N) | Longitude (°E) | Time period |
|---|---|---|---|---|
| *ALE* | *Alert* | *82.4900* | *-62.4200* | *2004-2013* |
| BON | Bondville | 40.0667 | -88.3667 | 1995-2019 |
| BOU | Boulder | 40.0500 | -105.0070 | 1992-2015 |
| CAB | Cabauw | 51.9711 | 4.9267 | 2013-2019 |
| DRA | Desert Rock | 36.6260 | -116.0180 | 1998-2019 |





| E13 | Southern Great Plains | 36.6050 | -97.4850 | 1994-2018 |
| FPE | Fort Peck | 48.3167 | -105.1000 | 1995-2019 |
| *GCR* | *Goodwin Creek* | *34.2547* | *-89.8729* | *1995-2019* |
| *GVN* | *Georg von Neumayer* | *-70.6500* | *-8.2500* | *1992-2019* |
| SPO | South Pole | -89.9830 | -24.7990 | 1992-2017 |
| SXF | Sioux Falls | 43.7300 | -96.6200 | 2003-2019 |
| SYO | Syowa | -69.0050 | 39.5890 | 1998-2019 |
| *TOR* | *Toravere* | *58.2540* | *26.4620* | *1999-2019* |


**Table 3: PROMICE sites serving as reference surface albedo data sources. Sites with less than 90% snow/ice cover in the containing CLARA grid cell are written in red italics.**

| Station | Elevation | Latitude (°N) | Longitude (°E) | Time period start |
|---|---|---|---|---|
| KPC_L | 370 | 79.9108 | -24.0828 | 17/07/2008 |
| KPC_U | 870 | 79.8347 | -25.1662 | 17/07/2008 |
| EGP | 2660 | 75.6247 | -35.9748 | 01/05/2016 |
| SCO_L | 460 | 72.223 | -26.8182 | 21/07/2008 |
| SCO_U | 970 | 72.3933 | -27.2333 | 21/07/2008 |
| *MIT* | *440* | *65.6922* | *-37.828* | *03/05/2009* |
| *TAS_L* | *250* | *65.6402* | *-38.8987* | *23/08/2007* |
| TAS_U | 570 | 65.6978 | -38.8668 | 15/08/2007 |
| TAS_A | 890 | 65.779 | -38.8995 | 28/08/2013 |
| *QAS_L* | *280* | *61.0308* | *-46.8493* | *24/08/2007* |
| *QAS_M* | *630* | *61.0998* | *-46.833* | *11/08/2016* |
| QAS_U | 900 | 61.1753 | -46.8195 | 07/08/2008 |
| QAS_A | 1000 | 61.243 | -46.7328 | 20/08/2012 |
| *NUK_L* | *530* | *64.4822* | *-49.5358* | *20/08/2007* |
| *NUK_U* | *1120* | *64.5108* | *-49.2692* | *20/08/2007* |
| *NUK_K* | *710* | *64.1623* | *-51.3587* | *28/07/2014* |
| *NUK_N* | *920* | *64.9452* | *-49.885* | *25/07/2010* |
| *KAN_B* | *350* | *67.1252* | *-50.1832* | *13/04/2011* |
| *KAN_L* | *670* | *67.0955* | *-49.9513* | *01/09/2008* |
| KAN_M | 1270 | 67.067 | -48.8355 | 02/09/2008 |
| KAN_U | 1840 | 67.0003 | -47.0253 | 04/04/2009 |
| *UPE_L* | *220* | *72.8932* | *-54.2955* | *17/08/2009* |
| *UPE_U* | *940* | *72.8878* | *-53.5783* | *17/08/2009* |
| *THU_L* | *570* | *76.3998* | *-68.2665* | *09/08/2010* |
| *THU_U* | *760* | *76.4197* | *-68.1463* | *09/08/2010* |
| CEN | 1880 | 77.1333 | -61.0333 | 23/05/2017 |


*Representativeness*

*In situ* measurements of surface albedo with precision radiometers with continuous maintenance and regular calibrations (as at e.g. the BSRN sites) offer the highest quality reference data source for validation of satellite-based albedo estimates.



However, the spatial footprint of these *in situ* measurements is of the order of tens to hundreds of meters, in stark contrast to the ~4 - 10 km spatial resolution of a single AVHRR-GAC image pixel, or the ~25 km resolution of the grid cells in the aggregated averages. This 'point-to-pixel' problem is a well-known challenge in the validation of satellite-based surface albedo as well as other surface parameters which can change rapidly in time and space (Wang et al., 2019). Without knowledge of how well or poorly the point-like measurement represents the area sampled by the satellite imager, it is difficult to assess whether agreements or disagreements between the satellite-based estimate and the reference measurement are indications of retrieval quality or simply manifestations of different measurement targets.

To investigate the impact of the spatial representativeness problem in the CLARA-A3 validation, we applied data from Google Earth Engine's Dynamic World (DW) dataset (Brown et al., 2022). Dynamic World provides continuously updated land cover data from Sentinel-2 at 10 m resolution at comparable accuracy to ESA World Cover data (Venter et al., 2022). We extracted areas corresponding with 0.25 degree / 25 km CLARA-A3 grid cells containing BSRN or PROMICE sites from DW. By applying climatological mean albedos for each land cover class after He et al. (2014) and Trlica et al. (2017), we obtained first-order estimates for the 'expected' mean surface albedo (BAL) in each grid cell in question through simple averaging (applied climatological albedos in Supplemental Material). While approximative, these estimates allow us to quantify the difference between the *in situ* measurement and its surrounding area, thus providing means to identify and exclude spatially unrepresentative sites from further analysis. Further, we would expect that the CLARA-A3 mean BAL should fall between the *in situ* and expected values, depending on the accuracy of DW classifications and the validity of the climatological mean albedo for the actual surface conditions in each classified DW pixel. Supplementary Figure S2 and associated text in Supplementary Material describe the results of this analysis for the non-polar BSRN sites.

Out of 13 non-coastal BSRN sites with long-term surface albedo measurements (

Table 2), we conservatively classified four (ALE, GCR, GVN, and TOR) as being unrepresentative at our resolution, excluding them from the results. Our selection is broadly similar with results from prior assessments of representativeness such as Liu et al. (2017).

For the PROMICE sites in Greenland, the question of point-to-pixel representativeness is complex. For sites close to the ice sheet margins, the surface conditions are notoriously heterogeneous even at short distances from the site (e.g. Ryan et al., 2017). Available means do not allow for robust matching of surface conditions at each PROMICE site's measurement footprint against grid cell (mean) conditions over the decadal span of the *in situ* data. We therefore elected to simply classify the PROMICE sites according to the coverage of snow and ice in their CLARA grid cell, omitting sites with <90% snow/ice coverage from summary statistics.

*Land*

With their multidecadal temporal coverage and regular on-site monitoring, the spatially representative BSRN sites are the principal reference data source in this study. To enhance the temporal representativeness aspect of the validation, each site's coordinates were tracked during CLARA-A3 processing and corresponding clear-sky level 2 (overpass) data was stored.





This allowed us to match clear-sky overpasses exclusively with *in situ* data from the same time periods (15-min windows), ensuring a direct comparison in the temporal domain. The level 2 data contain valid albedo estimates over snow-free land
surfaces for each individual overpass, thus allowing us to examine retrieval accuracy also in the GAC-resolution domain in addition to the grid cell domain whose spatial representativeness analysis was discussed above.

Figure 6 illustrates CLARA-A3 BAL bias over BSRN sites. Figure 6a shows MBE as a function of Sun Zenith Angle (SZA) of all matched snow-free level 2 data over the seven sites listed in the figure. A slight tendency to overestimate the in situ albedo at low SZA is contrasted by a similar underestimation at high SZA. The increasing error tendencies towards high
SZA justify the applied cut-off of 70 degrees for SZA in the satellite observations. Figure 6b illustrates the representativeness relationship in the observed bias; the more similar the grid cell land cover to that being measured at the corresponding BSRN site, the lower the estimation bias in general. Finally, Figure 6c shows the temporally resolved bias over each site, with annual and summer mean MBE printed for easier reference. Here, the evaluation includes both snow-covered and snow-free periods, explaining most of the rapid bias variations at sites like FPE which experience seasonal snow
cover.

Interestingly, at BOU and BON the summer mean biases are poor predictors of the annual bias. Wang et al. (2014) classified BON as poorly representative at the MODIS resolution level, which is consistent with the high annual bias seen here, although at our coarse grid cell scale the site appears more representative of its surroundings during summer. This variability leads us to conclude that representativeness should always be assessed at the specific resolution and grid cell extent being
investigated, and that the annual cycles of snow cover and vegetation phenology may produce markedly different estimation biases during different seasons.



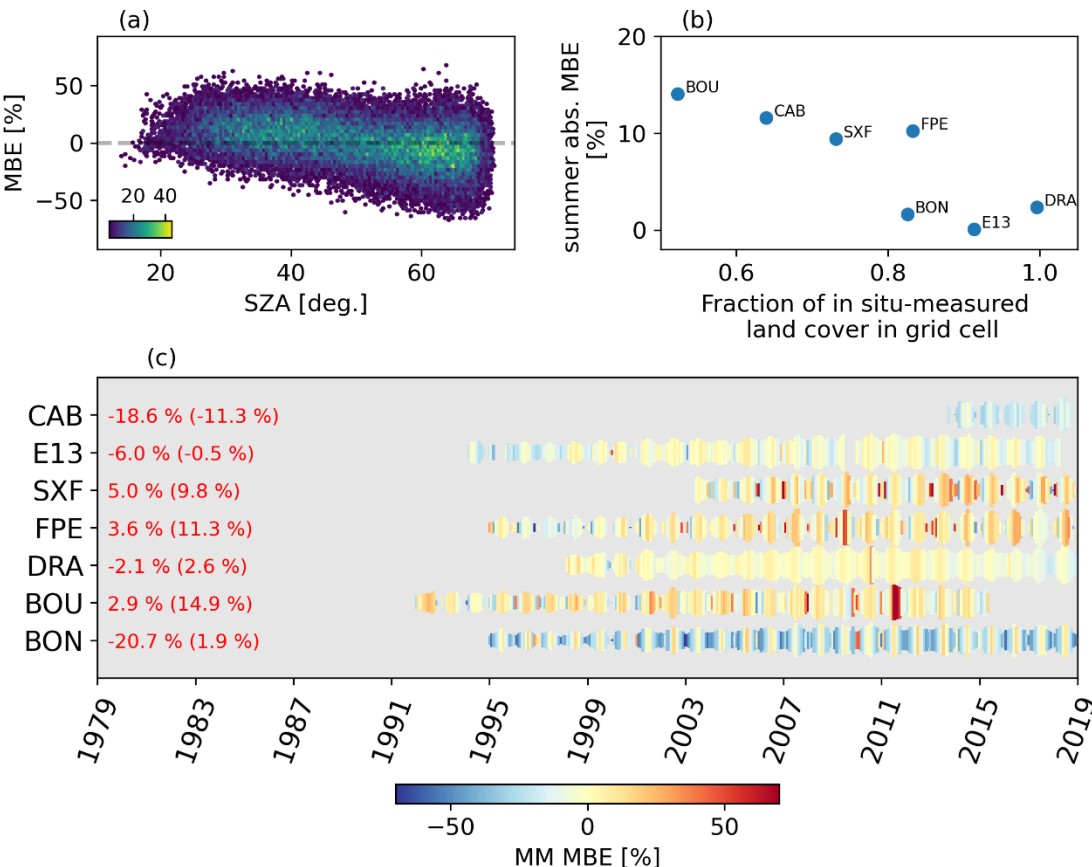

**Figure 6: (a) Relative retrieval error (MBE) as a function of SZA for all tracked level 2 snow-free blue-sky albedo (BAL) estimates**
**over the representative land surface BSRN sites (N=93620). Outliers with <5 occurrences are omitted for clarity. (b) Mean absolute MBE of summer (JJA) CLARA level 3 (i.e. 0.25 degree resolution) data as a function of fraction of land cover in the CLARA grid cell which matches the land cover being measured at the BSRN site. (c) Monthly mean MBE over the representative BSRN sites through time. Sizes of rectangular markers indicate the amount of valid clear-sky AVHRR data of each month. Text in red shows annual mean MBE (summer MBE).**

Figure 7 shows the site-averaged bias, precision, and stability for the 7 representative BSRN land sites, for all three albedo variables, at both temporal resolutions (pentad and monthly means). Performance in bias (Figure 7a) and stability (Figure 7c) is generally good, as seen in Figure 6. At CAB, the limited length of available in situ data (only about six years) likely affects both bias and stability estimates. Precision (Figure 7b), quantified through bias-corrected rms error, is where the 'legacy' nature of AVHRR as a sensor and retrieval algorithm limitations combine to produce a performance that is notably,

but not unexpectedly, inferior to data from more modern sensors such as MODIS (e.g. Wang et al., 2014). WAL precision tends to be lower than for SAL or BAL, also expectedly as WAL estimates are by nature derivatives of the clear-sky SAL retrievals, with typically higher and more variable biases. It should be noted that the in situ records are filtered separately for illumination conditions consistent with SAL, WAL, and BAL. Therefore, although BAL is a weighted mean of SAL and WAL, its metrics do not necessarily reflect a 'midpoint' of WAL and SAL performance.


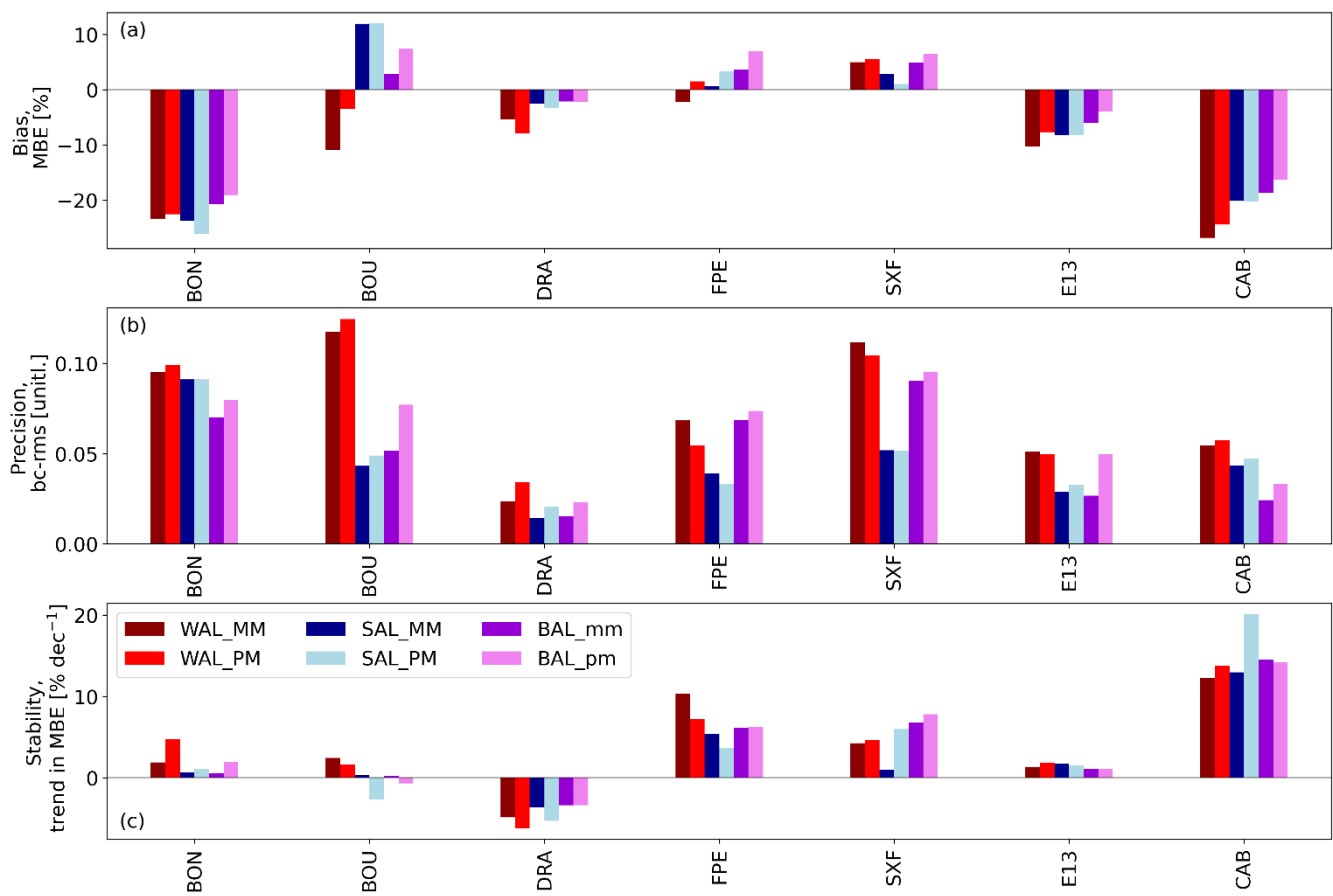

**Figure 7: Bias (MBE;panel a), precision (bc-rms; panel b) and stability (decadal trend in MBE; panel c) of the CLARA-A3 pentad and monthly mean albedo estimates over the representative BSRN land sites. Red for WAL, blue for SAL, and violet for BAL.**
**Dark colors indicate monthly means and light colors pentad means.**

Nevertheless, long-term biases are generally stable, implying that the algorithm itself is stable and the AVHRR radiance intercalibration efforts (Karlsson et al., 2023) have largely been successful during the evaluation period of 1995-2019 despite some issues in the latest years (details in Discussion). We next turn to the evaluation of retrieval accuracy over the cryospheric domain, which has been a prime application area for preceding CLARA albedo datasets.


*Greenland Ice Sheet and Antarctica*

Validation over the terrestrial cryospheric domain is focused on two *in situ* data sources: the PROMICE sites on the Greenland Ice Sheet, and SYO and SPO BSRN sites in Antarctica. Figure 8 illustrates the mean performance metrics over these sites for the temporally aggregated WAL, SAL, and BAL. The overall picture is similar to the preceding BSRN
analysis, with variable but generally low biases (Figure 8a) and good stability (Figure 8c). However, in precision (Figure 8b)



the performance is markedly lower regardless of the albedo variable being estimated, with the exceptions of sites deep within ice sheet interiors (CEN, EGP, KAN_U, SPO) where snow surface conditions have thus far remained primarily stable with minimal or no seasonal melt.

**Figure 8:** Bias (MBE; panel a), precision (bc-rms; panel b) and stability (decadal trend in MBE; panel c) of the CLARA-A3 pentad and monthly mean albedo estimates over the PROMICE and BSRN snow/ice sites. Red for WAL, blue for SAL, and violet for BAL. Dark colors indicate monthly means and light colors pentad means. Sites with less than 90% snow/ice cover in their CLARA grid cell are shown as partially transparent. SAL stability for EGP is not shown due to a very low number of available samples. The vertical grey line separates PROMICE and BSRN sites.

For stability, SYO and SPO stand out for specific reasons. At Syowa (SYO), located on East Ongul Island off the Antarctic mainland, notable breakups of land-fast ice in the area in 2016 (Aoki, 2017) and again in 2017 (Nakamura et al., 2022) led to underestimations of in situ albedo at the grid cell scale. While these disturbances are sufficient to produce a notable trend in bias, we note that it is not statistically significant at the 95% confidence interval. At South Pole (SPO), the diffuse/direct





illumination requirements for WAL/SAL matching led to very different samples of *in situ* observations, with WAL matchable only after the year 2000. As the measured albedo at the site had been notably low during 1995-1999 (~0.8, typical mean of ~0.85-0.9 afterwards), SAL had overestimated during the early part of the validation period. As this bias returned to low levels after 2000, a considerable negative trend in SAL bias was produced as a result.

*Arctic sea ice*

Over sea ice, we based the validation on reference data from two field campaigns which have provided *in situ* albedo measurements spanning a full Arctic summer season when satellite-based estimates are viable: SHEBA data covers Arctic summer 1998, and Tara Arctic observations cover the summer of 2007. The recently concluded Multidisciplinary drifting Observatory for the Study of Arctic Climate (MOSAiC) campaign is not considered here as its data is not yet available. The

observations from SHEBA and Tara were of different design; during SHEBA, albedo was measured on transects several hundred meters long, ensuring good spatial coverage but temporally available only every few days. During Tara, measurements were at a single site but with continuous temporal coverage throughout the summer. Both ice camps drifted across the Arctic Ocean during their duration, resulting in the need to continuously update the CLARA grid cell matchups with the reference data geolocation. As coincident surface albedo coverage from satellites is limited to a single summer for

both campaigns, we only evaluate bias and not precision or stability.

Figure 9 shows the in situ-measured surface albedos and their corresponding CLARA-A3 estimates from the relevant grid cells over the Arctic Ocean. As described earlier, for SHEBA (Figure 9a) the mapping is CLARA-based, with the transect locations during each CLARA pentad being matched with the grid cell containing them. For Tara (Figure 9b), the mapping is based on finding the CLARA grid cell containing the ice camp separately for each hourly observation. This difference

explains the enhanced smaller-scale variability in BAL estimates against Tara observations; the bias itself is of course also affected by the point-to-pixel evaluation itself. Nevertheless, mean bias appears low against both in situ data, although variability in bias is considerable owing to the point-to-pixel challenges combined with the highly dynamic surface conditions over sea ice in the melting season. The CLARA estimates display a mean variation range between black- and white-sky albedo of 0.05-0.06, in accordance with prior literature (Key et al., 2001). As expected, during the cloudy Arctic

Ocean summers the BAL estimates typically tend towards WAL rather than SAL.

The comparison shown focuses on pentad means which are more capable of tracking the progress of summer melt across the sea ice zone. A similar evaluation of the monthly mean against Tara observations (Supplementary Figure S3) shows the development of a marked underestimation (10-15%) during early summer as a result of the ice camp drifting into grid cells whose monthly mean conditions no longer matched local conditions at the measurement site. Likewise, the underestimation

of BAL against SHEBA albedo in August is a persistent feature since CLARA-A1, and most likely reflects point-to-pixel comparison issues given that e.g. cloud screening has been enhanced in the current albedo processing scheme.



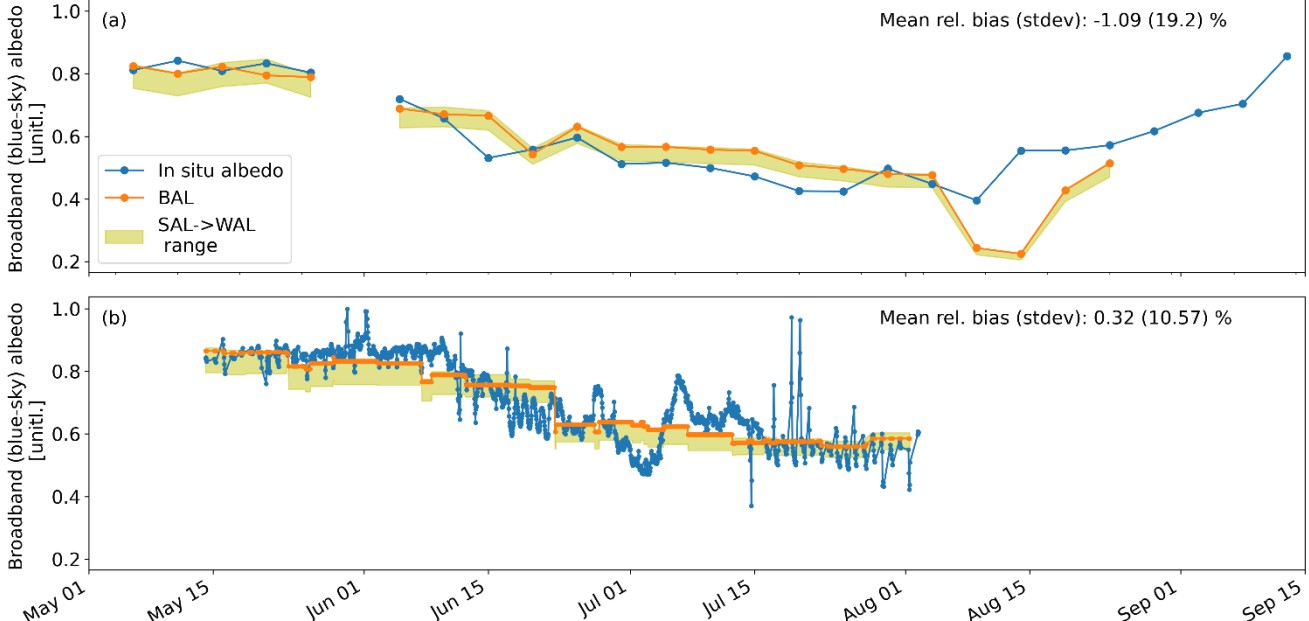

**Figure 9: In situ measured surface albedo of Arctic sea ice (blue) and the corresponding grid cell scale 5-day CLARA-A3 estimates (orange) for the (a) SHEBA expedition during summer 1998, and (b) for the Tara-Arctic expedition during summer 2007. The**
**yellow shading illustrates the range between the associated grid cell's black-sky and white-sky CLARA albedo estimates for each sampled period.**

Table 4 summarizes the results of the evaluation against *in situ* observations for the three albedo quantities. Regardless of applied reference data, bias is generally low (<10% relative) and its decadal stability is good (< 1 % dec.$^{-1}$) except for SAL and WAL against BSRN-Antarctica. Conversely, precision is low for both BSRN and PROMICE evaluations, although
likely with a substantial contribution from point-to-pixel comparison issues as discussed earlier. We note that SAL performance is fully comparable with the predecessor data records CLARA-A1 and CLARA-A2, with similar performance now found for the new WAL and BAL estimates.

**Table 4: Summary statistics for the evaluation categories and metrics of CLARA-A3 SAL / WAL / BAL (respectively). Values**
**shown are means over all valid reference data in each category.**

| Evaluation reference / time resolution | Bias [MBE, %] | Precision [bc-rms error, unitless] | Stability [trend of MBE, % dec.$^{-1}$] | N |
|---|---|---|---|---|
| BSRN-land / monthly | -4.5 / -9.6 / -4.2 | 0.043 / 0.072 / 0.063 | -0.46 / 0.03 / 0.49 | 1352 / 1268 / 1619 |
| BSRN-land / pentad | -3.9 / -7.6 / -2.0 | 0.044 / 0.074 / 0.072 | -0.57 / -0.05 / 0.40 | 7644 / 6853 / 9332 |
| PROMICE & | -5.05 / 3.86 / 0.95 | 0.123 / 0.157 / 0.115 | 0.00 / -0.60 / -0.33 | 538 / 643 / 624 |





| | | | |
|---|---|---|---|
| BSRN-Antarctica / monthly | | | |
| PROMICE & BSRN-Antarctica / pentad | -2.2 / 1.56 / 2.36 | 0.116 / 0.157 / 0.135 | 1.12 / -3.80 / -0.42 | 1716 / 1112 / 4050 |
| Arctic sea ice (SHEBA & Tara) / monthly | -3.59 | N/A | N/A | 8 (unique months) |
| Arctic sea ice (SHEBA & Tara) / pentad | -0.39 | N/A | N/A | 39 (unique pentads) |

## 4 Intercomparison to MODIS-based MCD43 and predecessor CLARA-A2 surface albedo data records

To place the newest CLARA CDR on surface albedo in context, we carried out an intercomparison between it and two other surface albedo data records; the MODIS-based MCD43 (Collection 6.1; Schaaf and Wang, 2021) and the predecessor

CLARA-A2 (Karlsson et al., 2017). The intercomparison considered black-sky albedo data between April – September 2015 from all sources. The MODIS data are provided daily at 0.01 degree resolution and normalized to local solar noon conditions, requiring preprocessing to match the coarser CLARA estimates. MODIS albedo estimates were first bucket-resampled to the 0.25 degree grid and aggregated into monthly means. Then both CLARA-A3 and MODIS aggregates were re-normalized to a common SZA of 60 degrees using GLOBCOVER land cover data and the equation of Briegleb et al.

460 (1986).

Figure 10 displays the intercomparison between MCD43 and CLARA-A3 as the period mean. In general, CLARA-A3 retrieves higher albedo over broadleaved (tropical) and deciduous forests, with croplands, grasslands and shrublands being very similar to MCD43. Conversely, MCD43 albedos over barren and desert regions are higher than for CLARA-A3. These differences are highly similar to the those observed when comparing CLARA-A1 with MCD43 (Riihelä et al., 2013). Given

the advances in CLARA retrievals and input data over the editions, it now seems likely that the differences are mainly attributable to core differences in the atmospheric and angular isotropy correction models used in these records. Indeed, the comparison between CLARA-A3 and its predecessor CLARA-A2 over the same period (Supplementary Figure S4) displays relatively close agreement, although we note that CLARA-A3 black-sky albedos over land and snow/ice surfaces are generally larger than in CLARA-A2. Particularly pre-melt Arctic sea ice and snow surfaces are now brighter by 0.02 – 0.05

on average. Interestingly, Antarctic sea ice is now dimmer in CLARA-A3, although the snow cover of Antarctica is slightly



brighter in A3 than in A2. Given that the retrieval gives equivalent treatment to Arctic and Antarctic sea ice, it is likely that either cloud screening or the updated atmospheric composition from ERA5 play a role in the change.

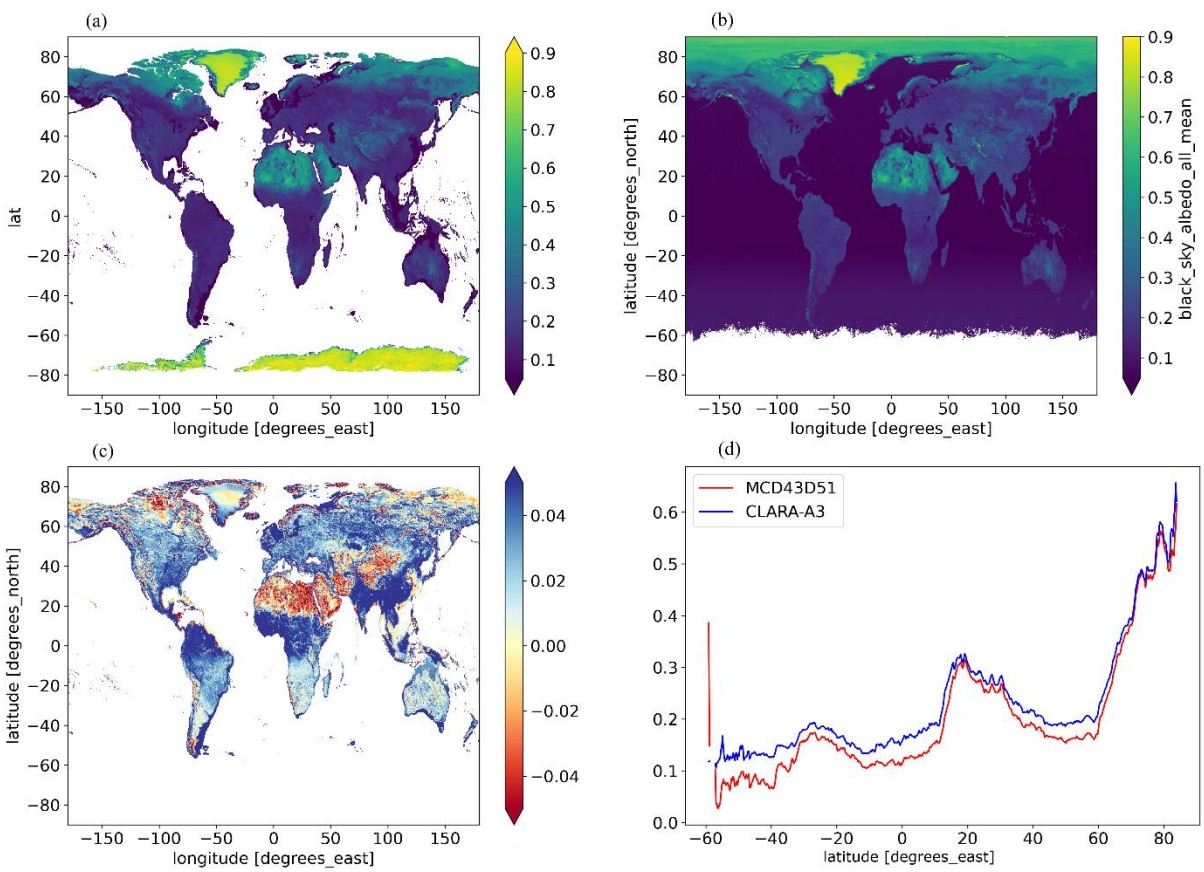

**Figure 10: Intercomparison of CLARA-A3 and MCD43D51 black-sky albedo estimates. Data shown corresponds to the mean of April-September 2015. (a) MCD43D51, (b) CLARA-A3, (c) difference (CLARA-MCD43), (d) zonal means of MCD43D51 and CLARA-A3.**

## 5 Discussion, strengths and limitations of the CLARA data

The surface albedo data record in CLARA-A3 has been shown to match or improve upon its predecessors in performance when evaluated against in situ observations or compared against other surface albedo data records. However, the data record also has its limitations. First, the relatively coarse spatiotemporal resolution requires careful consideration when applying the data to study any small-scale or rapid phenomena impacting albedo. Second, as discussed in Karlsson et al. (2023), the intercalibration accuracy for AVHRR radiances from the newest sensor-carrying satellites such as Metop-C is likely lower for the last years of the CDR (2019 & 2020). While the impact manifests only partially in the albedo estimates where all available AVHRR observations are always used in unison, users are advised to consider 2019 and 2020 as having larger than



normal retrieval uncertainty. Third, some periods of the record exhibit minor artefacts resulting from malformed AVHRR source data. During the early period of CDR (pre-1990), some individual grid cells poleward of the SZA cutoff contain albedo estimates, these result from incorrectly geolocated AVHRR observations being placed there. Similarly, grid cells at the dateline in a few of the North Polar albedo subsets during spring may show anomalous estimates, likely from erroneous geolocations in source AVHRR data.

For the accuracy of the albedo data record (and trends therein), it is crucial to base the retrievals on an accurately intercalibrated AVHRR radiance data record. The high decadal stability over validation sites in CLARA-A3 albedo (Figure 7 and Figure 8) suggests that the radiances, the atmospheric composition inputs, and the retrieval algorithm itself are now generally stable, although the issue is likely different over regions with e.g. high and variable aerosol loading (see Validation Report for details).

We may also examine the stability of the CDR time series through visualization of deseasonalized anomalies against a reference period (1982-1998). Figure 11 illustrates these zonal mean anomalies for black-sky albedo across the CLARA-A3 CDR coverage. The reference period and figure style are chosen to resemble Figure 11 of Riihelä et al. (2013) for easy comparison to CLARA-A1. Relative to CLARA-A1, the application of observation-based aerosol data and robust ERA5-based atmospheric composition in CLARA-A3 clearly reduce mid-latitude albedo anomalies, although the impacts of Mt.

Pinatubo eruption of 1992 remain only partly compensated for. Over the polar regions, the associated positive albedo anomalies in 1992 may be physically motivated, as the stratospheric cooling following the eruption may well have favored a cooler summer with inhibited surface melt, although assessing the atmospheric surface level response to volcanic excitation remains challenging due to the internal variability of climate models (Polvani et al., 2019). Post-2015, the lack of variability in mid-latitude anomalies reflect the use of an aerosol climatology, although the polar anomalies of this period are fully

consistent with observed gains and losses in polar snow and sea ice cover.

Earth System
Open Access    Science    Discussions
Data

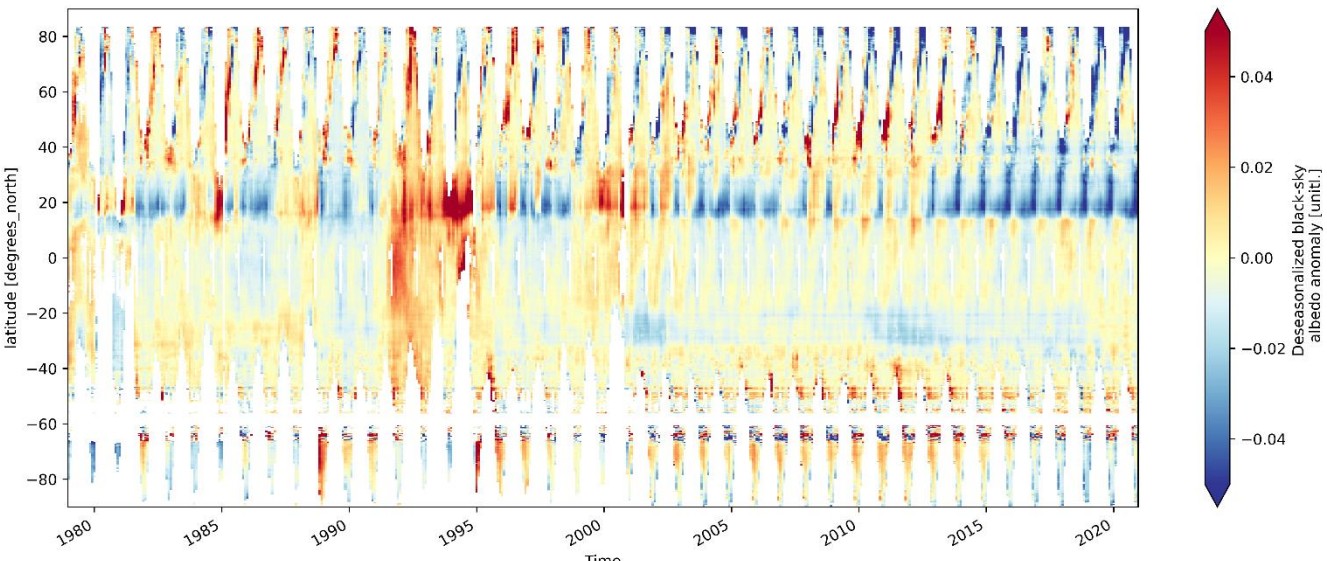

**Figure 11: Deseasonalized zonal monthly mean black-sky surface albedo anomalies against 1982-1998 mean. Data shown only for land & snow surfaces with >85% zonal coverage during the evaluated month.**

A principal design choice for CLARA albedo records has been the selection of 5 days of aggregated data (a pentad) as the shortest user-delivered temporal resolution. The implication here is that CLARA records cannot capture rapid surface albedo changes at shorter temporal scales. However, it has been shown that when aided with fitting functions, CLARA pentad data are sufficiently well-resolved to study e.g. melt onset and progress effects on snow albedo (e.g. Anttila et al., 2018; Kouki et al., 2019). Importantly, the increasing sampling density during the CDR period (Figure 4) likely enhances robustness in this

regard for the later years of the time series.

The predecessor CLARA surface albedo records have seen a notable uptake for cryospheric studies (e.g. Kashiwase et al., 2017; Karlsson and Svensson, 2013; Cao et al., 2015; Guo et al., 2018; Light et al., 2014; Thackeray and Hall, 2019; Zhang et al., 2019). With its extended coverage, demonstrated stability, and improved cloud detection, we expect that CLARA-A3 will continue to serve as a useful resource particularly for cryospheric investigations, with due attention given to its

limitations. Performance of the new white- and blue-sky albedo estimates is fully consistent with the 'core' black-sky albedo retrievals and should consequently broaden the array of potential application areas. Urraca et al. (2023) showed that the snow and ice albedo estimates in CLARA-A2 were closest to MODIS-based data algoritms are mature and calibration is well-known and stable, despite emergent calibration-related issues in the latest years of CLARA-A2. Given the updated calibration and continuity in the core retrieval algorithm from CLARA-A2 to A3, this finding further reinforces the belief

that CLARA-A3 will continue to prove of value for the cryospheric community in particular.

During the course of the data record preparation and evaluation, considerable attention was given to the performance assessment against in situ measurements, seeking to understand and explain the role that spatial representativeness (i.e. point-to-pixel problem) plays in the observed differences. Available evidence supports the view that large biases typically





result from poor comparability between the coarse-scale satellite estimate and the point-like in situ measurement, with the
'true' SAL algorithm uncertainty being likely 10 - 15% (relative) for typical atmospheric conditions. Although we
endeavoured to gather the majority of decade-spanning and robust in situ albedo measurements for this study, it should be
noted that emerging community-based validation tools like SALVAL (Sánchez-Zapero et al., 2023) could provide a future
platform for undertaking performance assessments with well-defined consistent procedures, metrics, and reference
observations for all participating data records, thus also facilitating their comparability.

**Data Availability**

The data record is distributed freely through the following DOI: 10.5676/EUM_SAF_CM/CLARA_AVHRR/V003
(Karlsson et al., 2023).

**Conclusions**

We have presented a new global surface albedo data record spanning over four decades. The data record is a component of
the third edition of the CLARA Climate Data Record family (CLARA-A3). It covers cryospheric, terrestrial and oceanic
domains and provides separate estimates for black-sky (SAL), white-sky (WAL) and blue-sky (BAL) surface albedo. The
spatiotemporal resolution remains the same as in the preceding CLARA editions. Alongside the albedo estimates, expanded
retrieval quality data is now provided to facilitate masking and screening as appropriate for each application.

We have undertaken a broad effort to evaluate the accuracy and stability of the data record against a selection of high-quality
*in situ* surface albedo measurements taken over terrestrial and cryospheric domains. From the results, we conclude that the
mean bias in CLARA-A3 SAL/WAL/BAL estimates is generally 10-15% (relative) when spatial representativeness issues
have been considered. Dedacal stability of said bias is also high ($< 2\%$ dec$^{-1}$), although the coarse spatiotemporal resolution
of CLARA does imply large scatter in retrieval errors across time and space (low precision). The observed performance
matches or improves upon the predecessor CLARA albedo records. Furthermore, we observe good agreement with
corresponding albedo data from MODIS-based MCD43 and the preceding CLARA-A2 records, though we note that
CLARA-A3 is typically somewhat 'brighter' than either of the other two. This is likely attributable to a combination of
higher base level in the new intercalibrated AVHRR radiance data, updated atmospheric composition data (from ERA5), and
a new cloud probability-based screening in CLARA-A3. An unexpected dimming in Antarctic sea ice albedo warrants
further study to determine whether it represents reality or undetected retrieval artefacts.
We expect that the third edition of CLARA surface albedo record, with expanded coverage and retrievals, will continue to
serve as a useful data source particularly for cryospheric studies. Together with the other components of CLARA-A3 which
describe global cloud and radiative energy parameters, the SAL, WAL, and BAL estimates will contribute to a more
complete understanding of the composition and evolution of the Earth's global energy budget.



## Appendices

### Author contribution

A.R. carried out the analyses and wrote the majority of the manuscript. E.J. prepared the AOD data record and contributed to the manuscript text. V. K-M. wrote the code for the validation of the CLARA-A3 albedo data and contributed to the manuscript text.

### Competing interests

The authors declare that they have no conflict of interest.

### Acknowledgments

The authors gratefully acknowledge the work of the CM SAF Operations Team during the preparation and processing of the CLARA-A3 data records, in particular Diana Stein at Deutscher Wetterdienst. The authors would also like to acknowledge the highly important work of Dr. Terhikki Manninen (retired) on the CLARA-A3 retrieval algorithms.

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
