# Peer review of "Four decades of global surface albedo estimates in the third edition of the CLARA climate data record"

_Earth System Science Data, 2023_

## Author Comment (AC1)

We would like to thank the reviewer for the careful assessment. Please find our replies below in red. Underlining highlights proposed additions, removals or revisions in manuscript.

The CLARA climate data record is the longest remote sensing based record for global surface variables, and has unique significance to environmental and climate studies. This manuscript presents an overview to the latest version of CLARA-A3 albedo product, including its algorithm update and features of dataset. The content is relevant to the scope of ESSD, and will be helpful to data users. The aim of this manuscript is to clearly describe the strength and limitations of the dataset. In this consideration, some questions and revision suggestions are raised as follows:

1. Firstly, I would complain that data is not directly accessable at the provided DOI: 5676/EUM_SAF_CM/CLARA_AVHRR/V003.

The data are orderable free of charge after registration in the CM SAF data portal. We (the CM SAF project) are required to keep registry and track usage of our data records, therefore direct downloads are not possible. The reviewer's listed DOI above seems to miss the "10." needed in the beginning, we checked that the DOI listed in the abstract and data availability section is complete and points to the correct web site of CM SAF.

2. In table 1, I just ask a question: is there any change in the algorithm of albedo retrieval over ocean surface?

The algorithm itself remains unchanged, but we have removed the normalization to 60 degrees SZA to maintain internal consistency in treatment of CLARA albedo estimates over all surfaces on Earth. Also, now the WAL and BAL estimates are also available over ocean surfaces, as noted in the table 1.

3. In Line 98, page 5, "As AVHRR geolocation is calculated on a geodesic reference ellipsoid, a combination of sufficiently large elevation and viewing angle requires across-track shifting of pixels to obtain true geolocation", I cannot easily understand the phase "requires across-track shifting of pixels to obtain true geolocation". And I think maybe the geometric processing of AVHRR data is not in the scope of this manuscript, otherwise, it would be too much to discuss in a single article.

The text refers to the need to alter the location of some pixels within the AVHRR swath if their true location (in mountainous terrain) differs too much from the original estimate, which is computed for flat terrain. For the reviewer's information, we repeat here Figure 3-5 from the data record ATBD, which illustrates the process; the geolocation estimate may differ markedly from true position in the across-track direction if the imaginary line from satellite to ground intersects true terrain position much 'earlier' than expected due to elevated terrain. This correction is applied only if the shift is larger than 0.5 times AVHRR resolution and is thus restricted to high mountains. We propose to add some clarifying text here, i.e. "As AVHRR geolocation is calculated on a geodesic reference ellipsoid (flat terrain), a combination of sufficiently…"

We nonetheless propose to keep the current brief description of topography correction, as it is one component of the CLARA SAL retrieval process. Detailed descriptions remain available in the ATBD. A similar procedure is described in Dech et al. (2020) for their AVHRR

processing, we will also refer to that paper for readers who desire additional references on the topic.

Dech S, Holzwarth S, Asam S, Andresen T, Bachmann M, Boettcher M, Dietz A, Eisfelder C, Frey C, Gesell G, et al. Potential and Challenges of Harmonizing 40 Years of AVHRR Data: The TIMELINE Experience. *Remote Sensing*. 2021; 13(18):3618. https://doi.org/10.3390/rs13183618

4. In Line 100, page 5, it is not clear whether the mean slope refers to slope of 30m pixel or 0.05DEG pixel. And whether the mean refers to the mean of 0.05DEG pixel or 0.25DEG pixel. If possible, please indicate the percentage of pixels which need topology correction.

Thank you, this was indeed not clearly presented and suffered from a typo. Specifically, the requirement is that the **maximum** slope between the 1/120 deg. GTOPO30 pixels contained in the GAC pixel under treatment must exceed 5 degrees in order for radiometric correction computations. The limit is exceeded only for mountainous areas (e.g. Rocky Mountains, Andes, Himalayas), but we do not have global statistics on the amount of retrievals being treated, as the condition is calculated for each AVHRR overpass separately. We propose revising the text here to "Over sufficiently rugged terrain (max GTOPO30 slope larger than 5 degrees in the GAC pixel, i.e. mountainous areas)".

5. In Line 103, page 5, it is not clear whether the "each AVHRR pixel to be corrected" refers to 05DEG pixel or 0.25DEG pixel.

The text refers to each GAC-resolution pixel, revised accordingly.

6. In line 123, page 6, it is not clear whether the same narrow-to-broadband conversion (NTBC) algorithm in Liang et al. (2000) is applied both for snow-ice surface and snow-free surface.

Line 120 attempted to make clear that the text refers to the snow-free terrain path. Revised the text for further clarity to "The narrow-to-broadband conversion (NTBC) algorithm for snow-free land also follows Liang..."

7. In line 231, page 9, "available sampling may change by a factor of 50", what is the meaning of "by a factor of 50"?

As seen in Figure 4b, by a factor of 50 means y = 50 * x; the available sampling for e.g. spring months in light color may be 50-100 GAC-resolution pixels, whereas in midsummer that number may reach 6000, thus 50-60 times larger.

8. In line 236, page 9, the "climatological albedo" is used in this manuscript as a reference data. But the source and accuracy of climatological albedo cannot be found throughout the manuscript.

We do not quite understand the comment. The climatological albedos listed here, with references to their original publications, are solely used in visualizing BAL variability about the expected value for (relatively) homogeneous terrain. The accuracy of these climatological estimates is of course no better than for climatologies in general, as both snow and vegetation are not static surface types. The intent here is simply to visualize the scatter and variability of CLARA BAL estimates against literature-based expectations for what the surface albedo would likely be for these types of terrain, assuming homogeneity.

9. Figure 4b, pages 10. I found it difficult to understand this subfigure. Are the scattered points in Figure 4b a single time sequence? If so, why are they not continuous as in Figure 4a?

Each marker indicates a single monthly mean over the limited region over central Greenland, whereas Fig 4a illustrates the global mean of available sampling per month. Due to this vast difference in sampled area, Fig 4b will naturally vary so much that a line plot would be quite hard to follow. It was also desired to encode the information on mean Solar Zenith Angle over the area into Fig 4b in order to prove that the variability is primarily related to lack of illumination in spring and fall, when the mean SZA approaches the cut-off limit of 70 degrees. If the reviewer deems it important, we can of course alter Fig 4a to also show a scatter plot-type of presentation, but using a continuous line plot for both subplots is not optimal in our view due to the reasons stated.

10. In line 284, page 12, "the decadal stability of bias, i.e. the temporal trend in bias as per cent per decade". The use of temporal trend as stability indicator is not questionable to me. It is possible the long-term trend is very small while the sequence

of albedo looks very noisy and unstable. I would suggest using variance of the bias as stability indicator.

The indicators chosen serve different purposes. The reviewer desires information about the variability of bias, which we present as the "Precision" indicator, with the bias-corrected rms error against reference as the metric. Being a rms metric, this indicator grows large in cases of noisy albedo retrievals as shown for some sites in e.g. Figures 7 and 8. The purpose of the "Stability" indicator is to provide information about the stability of the time series on annual and decadal scale to detect e.g. uncorrected radiometric calibration drifts of the AVHRR sensor constellation. These are important aspects to cover for considering the application of this time series to albedo trend detection. We thus argue that the provided indicators do provide information about the overall bias versus reference (bias), the noisiness of the retrievals (precision), as well as the long-term stability of the retrievals (stability).

11. Figure 6c, page 16, "Sizes of rectangular markers indicate the amount of valid clear-sky AVHRR data of each month." But I cannot find the rectangular markers in the figure.

Thank you, here perhaps the information was lost in the size of the multi-panel visualization. Each of the colored markers in Figure 6c is in fact a rectangle, whose size corresponds to the amount of data it is based on. The goal here is to not give excessive visual weight to large or small biases if they are based only on a very limited set of comparable data. To make this clearer to the reader, we propose revising the caption text to "The heights of the colored markers indicate…"

---

## Author Comment (AC2)

We would like to thank the reviewer for the careful consideration of the manuscript. Please find our responses to each raised issue below in red. Underlined text highlights proposed changes in the manuscript.

This manuscript discribed a long time series CLARA albedo climate data record from 1979 to present. This dataset is esential to environmental supervision and climate change studies. This study presents the overview of the retrieval algorithm and the validation of the new released CLARA albedo products. The topic is interesting to the the journal of ESSD. However, several issues existed and made the manuscript hard to understand. My major concerns are following:

- What's the main improvement of the new released albedo products? The high quality, the higher spatial resolution or the more frequency?

The temporal coverage is now extended backwards and forwards, covering 1979-2020, with the continuously produced interim data record on top. Some quality improvements were evident following updates in AVHRR radiance intercalibration and a new, improved cloud screening method based on probabistic calculations. Also, previously unused AVHRR instruments were included in the timeseries, increasing sampling frequency for some years of the early record. And finally, CLARA-A3 now includes black-, white-, and blue-sky albedo estimates for all surfaces for the first time in CLARA editions. We propose to revise the first paragraph of conclusions to briefly list these improvements.

- Up to now, I could not download your datasets (using the link as you provided DOI). How can you guarantee data access on the NTP data center servers?

We verified that the DOI-provided link to the CM SAF data portal did work as required. A (free) registration on the portal before data ordering is required, as CM SAF project is obligated to track data user statistics. In the future, the CLARA records may also become available through e.g. the Copernicus data provision services.

- I suggest separating the contents of "data record description" from the current of "Data record description and algorithm overview".

Very well, we will revise the manuscript to separate the sections so that data record description is its own section, followed by the algorithm overview.

- There are too much describtion about the albedo retrieval algorithm. I suggest the author make it more logical.

Given that the white- and blue-sky albedo estimates are now available for the first time, we feel that their algorithms must be presented in sufficient detail. However, for the black-sky albedo, we think that the ocean albedo description on lines 154-168 can be considerably condensed without loss of information to the reader. The other descriptions of e.g. treatment of snow and ice are in our view required, as cryospheric applications of the data record are a primary user interest.

- Does the retrieval algorithm have some difference with the proposed aogorithm?

We are unsure of what is meant here. The algorithm as described in the manuscript is the one applied consistently through all of CLARA-A3 albedo products.

- Does the NTB method can be directly used to transfer the narrowband reflectance to broadband albedo? As we know, the NTB method were built following the assumption of the land surface is Lambertian.

Over land surfaces, the BRDF correction first normalizes observed reflectances to nadir Sun/View geometry, and then the spectral albedo is derived from those reflectances following Roujean et al. (1992). These spectral albedo estimates are then used as input to the narrow-to-broadband conversion of Liang (2000).

While the NTBC algorithm was indeed based on Lambertian surface assumption, Liang argues in the reference paper that "this assumption was only for calculating the spectral distribution of downward flux. It is known that the spectral downward sky radiance and the integrated flux are not very sensitive to the anisotropy of surface reflectance (e.g., Liang & Lewis, 1996). Therefore, this assumption should not impact the derived formulae in this study." We thus assume that the resulting uncertainty is contained within the overall uncertainty in the second-order polynomial equation used for AVHRR – inspection of Fig 7 in Liang's paper suggests that uncertainties of 5-10% (relative) are expected for the NTB conversion.

- Which method was used to reduce the topographic effects on the imageries? Does the AVHRR imageries have the serious topographic effects at so coarse-scale of 0.25° to 25km?

The topography correction is calculated at the native GAC resolution of ~5 kilometers during the level 2 processing. At this level, we expect that steep mountain ranges such as Alps and Himalayas will produce noticeable effects in the AVHRR imagery.

- Which BRDF correction method was used in this manuscript?

This is stated on lines 120-121, "The BRDF correction and conversion to narrowband surface albedos for AVHRR CH1 and CH2 continue to follow the kernel-based approach of Wu et al. (1995) and Roujean et al. (1992)."

- Does any gap exist in the current new released CLARA albedo products follows your strict selection of observations (such as the Sun Zenith Angle >70 deg. or Viewing Zenith Angle >60 deg)?

Certainly. Polar regions are not covered from late autumn until mid-spring due to the SZA limitation. Also, during the earliest years when the AVHRR constellation was in fact only a single satellite at a time, the pentad means may have data gaps also at lower latitudes due to the VZA limitation. To make this clear to the reader, we propose adding a brief summary of data gaps to section 5 after line 515. Furthermore, we will note emerging EO-based techniques that could fill gaps due to clouds and lack of illumination in future CLARA editions, see Jääskeläinen et al., (2022).

Jääskeläinen, E., Manninen, T., Hakkarainen, J. and Tamminen, J.: Filling gaps of black-sky surface albedo of the Arctic sea ice using gradient boosting and brightness temperature data, International

Journal of Applied Earth Observation and Geoinformation, 107,102701 (2022).
doi:/10.1016/j.jag.2022.102701.

- How to screen out the snow-covered pixels for the retrieval of albedo over the global scale.

This is noted on lines 92-94, the snow covered area identification is handled during cloud parameter processing and is described in the given general reference paper of CLARA-A3.

- I can't understand the sentence "As AVHRR geolocation is calculated on a geodesic reference ellipsoid, a combination of sufficiently large elevation and viewing angle requires across-track shifting of pixels to obtain true geolocation" in Line 98-99 in page 5. Could you please explain it and make it more eaily to understand for the readers?

Thank you. This issue was also noted by reviewer #1, our response there contains the full explanation and a clarifying figure from the algorithm description document. Briefly, this refers to the fact that AVHRR geolocation is calculated over a smooth ellipsoid approximating the Earth's surface, which is not necessarily an accurate description over high-elevation areas. We correct for this when the assumption error is large, moving pixels within the AVHRR image. A similar approach for this type of correction is described by Dech et al. (2020) for their AVHRR processing. We will revise the text to make this clearer, also referring to the paper below.

Dech S, Holzwarth S, Asam S, Andresen T, Bachmann M, Boettcher M, Dietz A, Eisfelder C, Frey C, Gesell G, et al. Potential and Challenges of Harmonizing 40 Years of AVHRR Data: The TIMELINE Experience. *Remote Sensing*. 2021; 13(18):3618. https://doi.org/10.3390/rs13183618

- Does the SRTM have the DEM data between the surface that lies between the 56 degrees south to 60 degrees south latitude? As we all know, the SRTM's radars can cover most of Earth's land surface that lies between 60 degrees north and 56 degrees south latitude.

Thank you, this was indeed a mistake in the text. The SRTM coverage is 60 N – 56 S, with GTOPO30 being used elsewhere. We will revise the text accordingly.

- What is the meaning of the word of "TOMS" in line L111, page 5? I can't find any describtion about this word.

TOMS refers to the Total Ozone Mapping Spectrometer, a NASA satellite instrument series which provided measurements of ozone and aerosols between 1978 and 2006. We will revise the text to add the full name of the instrument.

- I find that you used the AOD data from 2015 to 2014 to replace the AOD data from 2005 to 2014. How many uncecessary uncertainties were introduced?

Yes, years 2015-2020+ are covered by a climatology because of emerging issues in our aerosol data source instrument OMI. While this certainly does introduce additional uncertainty, as noted on lines 117-119, it is difficult to quantify the exact additional

uncertainty. To obtain a general estimate for the reviewer, we obtained the CAMS monthly mean AOD550 data and calculated the differences to our AOD climatology for some example cases. Figures below illustrate the differences for April 2018 and July 2016. The data is resampled to the CAMS grid and resolution.

[Figure]

Larger differences occur over the boreal zone in summer, most likely with major contributions due to aerosols from forest and brush fires. Other large differences may occur over some tropical rain forest regions in Asia, as well as the Tibetan Plateau where CAMS AOD exhibits very small values, perhaps too small to be realistic.

To place these differences in context, we further did ATM simulations with the Py6S code (Wilson, 2013), as illustrated below. For an example case of (boreal) forests with a typical surface reflectance of 0.15 and viewing/illumination geometry matching common cases of AVHRR observation in summer, we find that variability of 0.3 in AOD can cause differences of up to 5% (relative) in surface reflectance for both AVHRR CH1 and CH2, with the broadband effect being in the same order of magnitude. We can therefore estimate that

typical additional relative uncertainty is likely in the 0-6% range, although larger effects remain of course possible.

Wilson, R. T., 2013, Py6S: A Python interface to the 6S radiative transfer model, *Computers and Geosciences*, 51, p166-171

[Figure]

[Figure]

- There are many Land Cover products, such as the USGS land cover products, the GLC2000 product, and so on. Do they have the same land cover classification principles? Do they have the same spatial resolution? If they have the unsame land cover classifications, how to merge them?

The different LC data are nearest neighbor-resampled to each AVHRR image within the limits of their resolution. However, it is important to note that in the albedo retrieval, land cover is treated only through coarse "archetypes" such as cropland, grassland, or barren. The various LC classes are mapped onto these archetypes with rules specific for each LC product. The procedure is space-consuming to explain in the manuscript, but the full details are available in the CLARA albedo Algorithm Theoretical Basis Document (ATBD), available freely through the CLARA-A3 DOI.

- Are the same albedo retrieval algorithms in the open wate and the closed water?

Yes, only one algorithm applies for all surfaces classified as water. Thus, large enough rivers do appear on the global surface albedo maps of CLARA.

- I can't understand the sentence "Note that the AVHRR-observed reflectances are not used in the estimation of ocean surface albedo." Why the AVHRR reflectance can not be used for generating land surface albedo?

Water albedo is determined to a large degree by the solar geometry and near-surface wind speed (waves, whitecaps), and its variability is less pronounced than for land surfaces. Also, given that the computational burden spared by treating ocean albedo through a model rather

than a full atmospheric correction-BRDF-NTBC albedo retrieval is very considerable over the decadal and global scale of CLARA, we have opted for the described processing choice.

- Does the statistics parameter used in Eq (2) and (3) can be used in the global scale?

Yes, the statistical relationships were derived from long timeseries of measured albedo data from all available BSRN sites with long-term coverage. While they represent the best-fit approximation, implying that cases with larger errors will occur, the underlying data mass is nevertheless large and our validation activities did not show any immediately clear cases of erroneous behavior at large scale.

- How to calculate the diffuse and direct radiation? How to calculate the CP? Which data can be used?

Cloud probability (CP) is calculated during the cloud parameter processing of CLARA-A3, which is a preceding step to the albedo estimation. The process is described in the CLARA-A3 overview paper of Karlsson et al. (2023), we will add the reference again to line 214 for clarity. Apart from this information and the solar zenith angle which comes from the satellite/solar geometry data, eq. 5 describes how direct radiation fraction is calculated, with diffuse radiation fraction being 1 – direct radiation.

- I suggest to more quantitative assessment of the new released CLARA albedo. The scatter plots can be used here both in the snow-covered surface and snow-free land surface.

We are unsure what is meant here. Figure 6a already shows the scatter plot of bias in all snow-free overpass-level BAL against the corresponding in situ measurements. The metrics in Table 4 are definitely quantitative for land, snow/ice and sea ice surfaces. If the reviewer wishes for a scatter plot of snow albedo bias, we will provide that for the pentads and monthly means as a supplementary figure.

- Which method was used to assess the representastiveness of the site?

This is explained on lines 301-332; we extract the site areas from high-resolution dynamic land cover data and assess its heterogeneity and resemblance between the measurement site coordinates and the 0.25 degree CLARA grid cell which contains it. For the ice sheet sites, we consider the coverage of ice in the CLARA grid cell as the metric for representativeness because of the large albedo difference between snow/firn/ice and snow-free terrain.

- I found that many sentences abou the quality of the new released albedos do not have the necessary figures or the tables to support, such as the sentence from 261 to 264, the sentence from 256 to 260, Line 265-274.

Very well, we will add supplementary figures containing the necessary reference data for skewness/kurtosis on large scale (first & second case), and the CDR-ICDR differences.

- How to select the cloud-free in situ albedo measurements to validation of the new released albedo product?

This is explained on lines 336-341; the validation site coordinates are tracked during processing, and if found in the overpass being processed, the relevant data are extracted and stored. This yields time series of clear-sky periods at each validation site.

- I suggest the author giving a scatter plot between the CLARA albedo and MCD43D51 in the global scale to show the accuracy of the CLARA albedo.

Very well, we will amend Figure 10 with the scatter plot.

- I found that the larger biases exist between the CLARA albedo and MCD43D51 albedo, especially over theSouthern Hemisphere. What are reasons?

Largest differences between CLARA and MCD43 occur over tropical forests (evergreen broadleaf). This has consistently been the case since the first CLARA release, as seen in Figure 9 of Riihelä et al. (2013). Alongside differences in BRDF modeling, high aerosol loading and subpixel cloud contamination in the coarser AVHRR imagery have been proposed as explanations for the behavior. Desert regions are the other notably different region, likely also related to aerosol loading and surface BRDF treatment.

- The DPI fo the figures need to be improved.

The figures were uploaded at 300DPI as per journal guidelines. We of course have higher-resolution versions available as needed.

---

## Author Comment (AC3)

We wish to thank the reviewer for the careful examination of the manuscript. Please find our responses below in red, with proposed amendments or revisions highlighted with underlined text.

**Minor issues:**

Method: the retrieval method is well known to be robust to generate good retrievals but it does not account for coupling between the surface and the scattering atmosphere close to it. Could the authors expand a bit on this point?

We assume that the comment refers to the simplified treatment for multiple scattering processes within the SMAC atmospheric correction algorithm employed in CLARA albedo retrieval. It is certainly true that multiple scattering is treated simply through the spherical albedo of the atmosphere (so not fully decoupled either), and an explicit multiple-layer atmosphere is not a component of the SMAC calculations; these were intentional original design choices to enable the very fast correction calculations in SMAC which are inherently necessary for the very large data volumes present in the 40-year CLARA timeseries (Rahman and Dedieu, 1994).

Rahman, H., & Dedieu, G. (1994). SMAC: a simplified method for the atmospheric correction of satellite measurements in the solar spectrum. Remote Sensing, 15(1), 123-143.

It is difficult to estimate the degree of additional error resulting from the simplicity of the multiple interaction accounting, given the overall simplicity of the SMAC approach. Certainly, we know that the additional error is a function of viewing and illumination geometry, which is yet another reason for our relatively conservative angular cutoffs in processing when compared to e.g. MODIS albedo products. We propose revising the text around line 110 to include a cautionary note on the underlying limitations in SMAC, noting that the effects will manifest mostly in cases of low Sun and/or satellite zenith angles.

Explain the meaning of the acronyms SAL, WAL, BAL (pag 3 line 63). The three albedos are explained with equations in the relative paragraph, but it is missing a sentence explaining them. Those variables are just not only mathematical terms but have a meaning and I think this must be added, in particular for non-expert readers or data users.

We propose to expand the text around line 63 as follows to explain the physical interpretation of the albedo quantities, particularly for non-expert readers:

"Conceptually, black-sky albedo would be observable in the absence of an atmosphere, when all solar illumination comes from a single direction. Conversely, white-sky albedo would be observable only in cases where the incoming illumination is fully diffuse, i.e. evenly distributed from all directions in the sky. In real-world situations on Earth, neither extreme case is achievable, and the incoming illumination is a combination of direct and diffuse radiation fluxes. The blue-sky albedo is the parameter that seeks to estimate these cases."

Table 1: add reference for ERA-Interim and ERA5.

Revised as requested.

Pag. 4 Add a similar plot (Figure 1 a) for a pentad product. This is necessary to give a complete example of the different products provided to the users. This would show if specific difference can be due to the amount of days accumulated.

We will revise Figure 1 to include the map and zonal means of monthly example as a/b, and map plus zonal means of one pentad within that month as c/d.

Pag. 12, Table 2 add a column to define the surface type of each site. It would be of great help for the readers to have a plot showing the location on Earth of the sites.

We will add the land covers of the sites to the table. We will also add the site locations as markers to Figure 1a.

Pag. 24, Figure 11. This figure has potential for me but the re no clear meaning on what is the message in it. Could the authors spend a couple of sentences to add a deeper explanations?

Figure 11 is meant as a comparison to a similar figure produced for the publication describing the first CLARA edition (Riihelä et al., 2013). The underlying message is rather simple, the colors indicate zonal mean anomalies against 1992-1998 mean albedo (of land and snow) for each month and latitude in the CLARA-A3 time series. While in general low values are desirable particularly for the low latitudes where snow is not present and land surfaces change relatively slowly, the figure illustrates well e.g. the impact of the Mt. Pinatubo eruption in 1992 as rapid anomalies in the albedo data, partly natural and partly due to imperfect atmospheric descriptions following this dramatic event.

Another message to be obtained here is the consistently negative anomaly over the "desert belt" of 20 N latitude for 2014-2020, most likely following from the use of the AOD climatology which would mask some of the natural variability seen during the prior decades of the record. Finally, the anomalies are generally now lower than for CLARA-A1, demonstrating that the retrievals are more stable due to advances in algorithms and supporting data, such as cloud screening. We will add these notes to the text.

**Major issues:**

Pag 3 line 62,63: You introduce the white, black, blue sky albedos. You miss to explain their meaning and their relation. This should be add. See for instance (https://doi.org/10.1175/JAS3479.1)

The comment appears to refer to the same item seen above in minor comments; please see the proposed revision there for additional information to the reader.

Pag. 10 Figure 4. I suggest to add a figure similar to (b) but for one region over the tropics. This will show how different geographical regions are impacted in the retrieval and provide the users with a clear view of the potentialities of the products in different areas.

We propose to add a subplot into Figure 4, illustrating sampling over e.g. the rainforests of central Africa as a counterpoint to the polar region of subplot b.

Pag. 9 Line 219. The authors say that being the retrieval deterministic rather than probabilistic but the measure of uncertainty is left to statistical values. The albedo variables are estimated using clear mathematical relations. Could the authors explain what the uncertainty are not also mathematically estimated using the well known procedures. See for instance the method used in https://doi.org/10.1029/2006JD007313. I find this section personally to be the only one a bit weak in the paper, otherwise well structured.

The formalism in the paper referenced (Govaerts & Lattanzio, 2007) is designed for geostationary imagers. A central idea there is that since the viewing geometry is fixed, variability in derived reflectivity shall correlate with errors in the instrument, the radiative transfer calculus or the atmospheric description relevant for the scattering/absorption processes along the pathway. The continuously variable solar/viewing geometry of AVHRR and the diurnally limited sampling of any one place on the Earth would imply violation of those assumptions if we were to attempt implementing the same approach for CLARA error estimation.

The geostationary viewpoint also allows for sub-daily characterization of e.g. AOD variability for the uncertainty characterization, which is not possible for polar orbiters (unless the imager constellation is very large). Finally, the method assumes non-variability of the surface albedo over an extended sampling period to maximize the robustness of the uncertainty estimate. While this is certainly achievable and a valid assumption for low-latitude vegetated surfaces, high-latitude snow and ice cover will cause sudden dynamic changes in reflectivity that would invalidate uncertainty estimates using this approach.

This being said, we are not at all opposed towards improved characterizations of uncertainty for the CLARA retrievals. One approach being considered is to use the level 2 validation data gathered during the CLARA-A3 production process as a source for training machine learning methods to create a retrieval error prediction algorithm for the next edition, that is foreseen to keep the present core retrieval algorithms. It could also be possible to release separate per-pixel uncertainties for CLARA-A3 *a posteriori*, although this would depend on the robustness of the machine learning-based uncertainty estimates, given that the available (reference) validation data is spatially limited on the global scale.

Section 3 on validation. The authors have a very long list of validation sites. Even if they clearly worked a lot and well on this part, I have several issues with this paragraph. The authors includes a lot of sites (in red italics) not used in the analysis being "spatially" unrepresentative. Could you explain why they have been included if not sued? For the remaining sites, I would have expected some full time range (4 decades or the time covered by the ground measurements at least) plots (selecting some representative among all, covering the different surface types). Only basing the analysis on monthly mean could be misleading for me. There are some summary statistics in Table 4. The table shows interesting figures. I would recommend to provide the numbers for each valid site and not as merged statistic.

Thank you for the comment, we considered this carefully. We propose to first remove the unused sites from tables 2 and 3, but we note the excluded sites in the text regarding the representativeness analysis for clarity. Second, we propose to add a selection of time series plots showing the CLARA estimates and the reference albedos, in style of Figure 9. The question of how to clearly illustrate both monthly and pentad data over decadal scale has been considered for each CLARA release validation, and unfortunately no optimal choice has

been found. We suggest showing two sites from BSRN and two sites from PROMICE, with separate subplots for pentads and monthly means vs. reference to illustrate the variability. This implies 8 panels, which is already a large figure, but experience has shown that mixing monthly and pentad data into the same subplots degrades readability considerably, as unfortunately does portrayal of multi-decadal data in a single plot.

Regarding the site-specific results, Figures 7 and 8 are specifically designed to illustrate the bias and other quality indicators on a site-to-site basis. We thus suggest not to add additional large tables to the already rather lengthy manuscript.

Section 4 on intercomparison to MODIS. This is the only comparison against satellite data. It is a pity not to intercompare against a geostationary product but I can see the difficulty even if it would have made the section more complete. The comparison against MODIS is a bit weak. There is only one figure, again showing means over months. It would be really important to select some ground sites and show time series including also MODIS and CLARA-3.

We will expand the visualization here to include a time series-based comparison of MCD43 and CLARA over a small selection of ground sites.